# Task-Agnostic Machine-Learning-Assisted Inference

**Jiacheng Miao**
University of Wisconsin-Madison
jiacheng.miao@wisc.edu

**Qiongshi Lu**
University of Wisconsin-Madison
qlu@biostat.wisc.edu

## Abstract

Machine learning (ML) is playing an increasingly important role in scientific research. In conjunction with classical statistical approaches, ML-assisted analytical strategies have shown great promise in accelerating research findings. This has also opened a whole field of methodological research focusing on integrative approaches that leverage both ML and statistics to tackle data science challenges. One type of study that has quickly gained popularity employs ML to predict unobserved outcomes in massive samples, and then uses predicted outcomes in downstream statistical inference. However, existing methods designed to ensure the validity of this type of post-prediction inference are limited to very basic tasks such as linear regression analysis. This is because any extension of these approaches to new, more sophisticated statistical tasks requires task-specific algebraic derivations and software implementations, which ignores the massive library of existing software tools already developed for the same scientific problem given observed data. This severely constrains the scope of application for post-prediction inference. To address this challenge, we introduce a novel statistical framework named PSPS for task-agnostic ML-assisted inference. It provides a post-prediction inference solution that can be easily plugged into almost any established data analysis routines. It delivers valid and efficient inference that is robust to arbitrary choice of ML model, allowing nearly all existing statistical frameworks to be incorporated into the analysis of ML-predicted data. Through extensive experiments, we showcase our method's validity, versatility, and superiority compared to existing approaches. Our software is available at `https://github.com/qlu-lab/psps`.

## 1 Introduction

Leveraging machine learning (ML) techniques to enhance and accelerate research has become increasingly popular in many scientific disciplines [44]. For example, sophisticated deep learning models have achieved remarkable success in predicting protein structure and interactions, which has the potential to significantly speed up the research process, save costs, and revolutionize the field of structural biology [1, 2, 25]. However, recent studies have pointed out that statistical inference using ML-predicted outcomes may lead to invalid scientific discoveries due to the lack of consideration of ML prediction uncertainty in traditional statistical approaches. To address this, researchers have introduced methods that couple extensive ML predictions with limited gold-standard data to ensure the validity of ML-assisted statistical inference [3, 35, 46].

Despite these advances, current ML-assisted inference methods can only address very basic statistical tasks, including mean estimation, quantile estimation, and linear and logistic regression [3, 35]. While the same mathematical principle behind existing ML-assisted inference methods can be generalized to a broader class of M-estimation problems, specific algebraic derivations and computational implementations are required for each new statistical task. Moreover, many tasks, such as the Wilcoxon rank-sum test, do not fit into the M-estimation framework. These issues pose significant challenges to the broad application of ML-assisted inference across various scientific domains.

38th Conference on Neural Information Processing Systems (NeurIPS 2024).

Historically, the field of statistics has faced similar types of challenges. Before the advent of resampling-based methods [19], it used to require task-specific derivation and implementation to obtain the variance of any new estimator. This old problem mirrors the current state of ML-assisted inference, where every new task requires non-trivial effort from researchers. However, with resampling-based inference, the need to manually derive variance is reduced. Instead, resampling methods can be universally applied to many estimation problems and easily obtain variance [17–19]. Inspired by this, we seek a universal approach that incorporates ML-predicted data into any existing data analysis routines while ensuring valid inference results.

We introduce a simple protocol named **P**o**S**t-**P**rediction **S**ummary-statistics-based (PSPS) inference (Fig. 1). It employs existing analysis routines to generate summary statistics sufficient for ML-assisted inference, and then produces valid and powerful inference results using these statistics. It has several key features:

- *Assumption-lean and data-adaptive*: It inherits the theoretical guarantees of validity and efficiency from state-of-the-art ML-assisted inference methods [4, 20, 35]. These guarantees hold with arbitrary ML predictions.

- *Task-agnostic and simple*: Since our method only requires summary statistics from existing analysis routines, it can be easily adapted for many statistical tasks currently unavailable or difficult to implement in ML-assisted inference.

- *Federated data analysis*: It does not need any individual-level data as input. Sharing of privacy-preserving summary statistics is sufficient for real-world scientific collaboration.

## 2 Problem formulations

### 2.1 Setting

We focus on statistical inference problems for the parameter $\theta^* \equiv \theta^*(\mathbb{P}) \in \mathbb{R}^K$ defined on the joint distribution of $(\mathbf{X}, Y) \sim \mathbb{P}$, where $Y \in \mathcal{Y}$ is a scalar outcome and $\mathbf{X} \in \mathcal{X}$ be a $K$-dimensional vector representing features. We are interested in estimating $\theta^*$ using labeled data $\mathcal{L} = \{(\mathbf{X}_i, Y_i), i = 1, \cdots, n\} \equiv (\mathbf{X}_\mathcal{L}, Y_\mathcal{L})$, unlabeled data $\mathcal{U} = \{\mathbf{X}_i, i = n+1, \cdots, n+N\} \equiv \mathbf{X}_\mathcal{U}$, and a pre-trained ML model $\widehat{f}(\cdot) : \mathcal{X} \to \mathcal{Y}$. Here, $f(\cdot)$ is a black-box function with unknown operating characteristics and can be mis-specified. We also require an algorithm $\mathcal{A}$ that inputs the labeled data $\mathcal{L}$ and returns a consistent and asymptotically normally distributed estimator $\widehat{\theta}$ for $\theta^*$. There are three common ways in the literature to estimate $\theta^*$:

- **Classical statistical methods** apply algorithm $\mathcal{A}$ to only labeled data $\mathcal{L} = (\mathbf{X}_\mathcal{L}, Y_\mathcal{L})$, and returns the estimator and its estimated variance $[\widehat{\boldsymbol{\theta}}_\mathcal{L}, \widehat{\mathrm{Var}}(\widehat{\boldsymbol{\theta}}_\mathcal{L})]$. Valid confidence intervals and hypothesis tests can then be constructed using the asymptotic distribution of the estimator. However, it ignores the unlabeled data and ML prediction.

- **Imputation-based methods** treat ML prediction $\widehat{f}$ in the unlabeled data as the observed outcome, and apply algorithm $\mathcal{A}$ to $\mathcal{U} = (\mathbf{X}_\mathcal{U}, \widehat{f}_\mathcal{U})$. We denote the estimator and estimated variance as $[\widehat{\boldsymbol{\eta}}_\mathcal{U}, \widehat{\mathrm{Var}}(\widehat{\boldsymbol{\eta}}_\mathcal{U})]$. This has been shown to give invalid inference results and false scientific findings [3, 35, 36, 46].

- **ML-assisted inference methods** use both $\mathcal{L}$ and $\mathcal{U}$ as input. These approaches add a debiasing term in the loss function (or estimating equation) for M-estimation problems, thus removing the bias from the imputation-based estimators and producing results that are statistically valid and universally more powerful compared to classical methods [4, 35, 36].

Next, we use an example to provide intuition on ML-assisted inference and our protocol.

### 2.2 Building the intuition with mean estimation

We consider the mean estimation problem, where $\theta^* = \mathbb{E}[Y_i] \equiv \arg\min_\theta \mathbb{E}[\frac{1}{2}(Y_i - \theta)^2]$. The classical method only takes the labeled data $Y_\mathcal{L}$ as input and yields an unbiased and consistent estimator for $\theta^*$: $\widehat{\theta}_\mathcal{L} = \arg\min_\theta \frac{1}{n} \sum_{i=1}^{n} \frac{1}{2}(Y_i - \theta)^2 = \frac{1}{n} \sum_{i=1}^{n} Y_i$. The imputation-based method only takes the unlabeled data $\widehat{f}_\mathcal{U}$ as input and returns $\widehat{\eta}_\mathcal{U} = \arg\min_\theta \frac{1}{N} \sum_{i=n+1}^{n+N} \frac{1}{2}(\widehat{f}_i - \theta)^2 = \frac{1}{N} \sum_{i=n+1}^{n+N} \widehat{f}_i$. It is a biased and inconsistent estimator for $\mathbb{E}[Y_i]$ if the ML model $\widehat{f}$ is mis-specified.

To address this, ML-assisted estimator takes both labeled data $(Y_{\mathcal{L}}, \widehat{f}_{\mathcal{L}})$ and unlabeled data $\widehat{f}_{\mathcal{U}}$ as input and adds a debiasing term to the loss function to rectify the bias caused by ML imputation [3, 20, 35, 36]:

$$\widehat{\theta}_{\texttt{MLA}} = \arg\min_\theta \frac{1}{2}\{\widehat{\omega}_0 \frac{1}{N}\sum_{i=n+1}^{n+N}(\widehat{f}_i - \theta)^2 - \underbrace{[\widehat{\omega}_0 \frac{1}{n}\sum_{i=1}^{n}(\widehat{f}_i - \theta)^2 - \frac{1}{n}\sum_{i=1}^{n}(Y_i - \theta)^2]}_{\text{Debiasing term}}\}$$

$$= \widehat{\omega}_0 \frac{1}{N}\sum_{i=n+1}^{n+N}\widehat{f}_i - \underbrace{[\widehat{\omega}_0 \frac{1}{n}\sum_{i=1}^{n}\widehat{f}_i - \frac{1}{n}\sum_{i=1}^{n}y_i]}_{\text{Debiasing term}},$$

where the modified loss ensures the consistency of the ML-assisted estimator and the weight $\widehat{\omega}_0 = \frac{\widehat{\text{Cov}}_l[Y,\widehat{f}]/n}{\widehat{\text{Var}}_l[\widehat{f}]/n + \widehat{\text{Var}}_u[\widehat{f}]/N}$ ensures that ML-assisted estimator is no less efficient than the classical estimator with arbitrary ML predictions: $\text{Var}(\widehat{\theta}_{\texttt{MLA}}) = \text{Var}(\widehat{\theta}_{\mathcal{L}}) - \frac{\text{Cov}[Y,\widehat{f}]}{n\,\text{Var}[\widehat{f}] + n^2\,\text{Var}[\widehat{f}]/N} \leq \text{Var}(\widehat{\theta}_{\mathcal{L}})$.

Our proposed method is motivated by the observation that the **sufficient statistics** of the ML-assisted estimator $\widehat{\theta}_{\texttt{MLA}}$ and its estimated variance $\widehat{\text{Var}}(\widehat{\theta}_{\texttt{MLA}})$ are the following summary statistics:

$$\widehat{\boldsymbol{\theta}}_{\texttt{ss}} = (\frac{1}{n}\sum_{i=1}^{n}y_i, \frac{1}{n}\sum_{i=1}^{n}\widehat{f}_i, \frac{1}{N}\sum_{i=n+1}^{n+N}\widehat{f}_i) \text{ and } \widehat{\text{Var}}(\widehat{\boldsymbol{\theta}}_{\texttt{ss}}) = \begin{bmatrix} \widehat{\text{Var}}_l[Y]/n & \widehat{\text{Cov}}_l[Y,\widehat{f}]/n & 0 \\ \widehat{\text{Cov}}_l[Y,\widehat{f}]/n & \widehat{\text{Var}}_l[\widehat{f}]/n & 0 \\ 0 & 0 & \widehat{\text{Var}}_u[\widehat{f}]/N \end{bmatrix}$$

Moreover, they can be easily obtained by applying the **same algorithm** $\mathcal{A}$ (mean estimation) to

- labeled data with observed outcome $\mathcal{A}(Y_{\mathcal{L}}) \to [\widehat{\theta}_{\mathcal{L}}, \widehat{\text{Var}}(\widehat{\theta}_{\mathcal{L}})] = [\frac{1}{n}\sum_{i=1}^{n}y_i, \widehat{\text{Var}}_l[Y]/n]$

- labeled data with predicted outcome $\mathcal{A}(\widehat{f}_{\mathcal{L}}) \to [\widehat{\eta}_{\mathcal{L}}, \widehat{\text{Var}}(\widehat{\eta}_{\mathcal{L}})] = [\frac{1}{n}\sum_{i=1}^{n}\widehat{f}_i, \widehat{\text{Var}}_l[\widehat{f}]/n]$

- unlabeled data with predicted outcome $\mathcal{A}(\widehat{f}_{\mathcal{U}}) \to [\widehat{\eta}_{\mathcal{U}}, \widehat{\text{Var}}(\widehat{\eta}_{\mathcal{U}})] = [\frac{1}{N}\sum_{i=n+1}^{n+N}\widehat{f}_i, \widehat{\text{Var}}_u[\widehat{f}]/N]$

- bootstrap of labeled data $\mathcal{A}[(Y_{\mathcal{L}}, \widehat{f}_{\mathcal{L}})_q, q = 1,\ldots,Q]$ for estimation of $\widehat{\text{Cov}}(\widehat{\theta}_{\mathcal{L}}, \widehat{\eta}_{\mathcal{L}}) = \widehat{\text{Cov}}_l[Y,\widehat{f}]/n$. Here, $(Y_{\mathcal{L}}, \widehat{f}_{\mathcal{L}})_q$ represents the $q$-th bootstrap of labeled data.

Combining these summary statistics for one-step debiasing $\widehat{\omega}_0\widehat{\eta}_{\mathcal{U}} - (\widehat{\omega}_0\widehat{\eta}_{\mathcal{L}} - \widehat{\theta}_{\mathcal{L}})$ recovers $\widehat{\theta}_{\texttt{MLA}}$.

To summarize, an algorithm for mean estimation, coupled with resampling, is sufficient for ML-assisted mean estimation. This observation inspired us to generalize this protocol for a broad range of tasks. Our protocol illustrated in Fig. 1 only requires three steps: 1) using a pre-trained ML model to predict outcomes for labeled and unlabeled data, 2) applying existing analysis routines to generate summary statistics, and 3) using these statistics in a debiasing procedure to produce statistically valid results in ML-assisted inference.

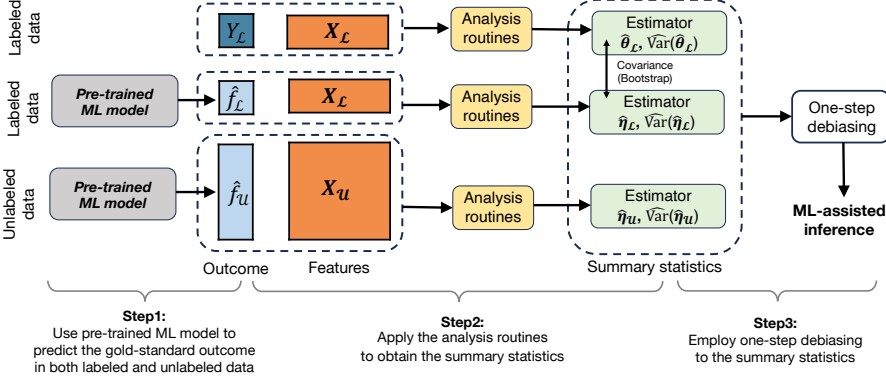

Figure 1: Workflow of PSPS for Task-Agnostic ML-Assisted Inference.

## 2.3 Related work

Our work is closely related to recent methods developed in the literature of ML-assisted inference [3, 4, 20, 35, 37, 46, 56], and is also related to methods for handling missing data [40, 42] and semi-supervised inference [6, 16, 50, 52]. While current ML-assisted inference methods modify the loss function or the estimating equation, our protocol works directly on the summary statistics. For simple problems such as mean estimation, current methods yield a closed-form solution to the optimization problem. However, for more general statistical tasks, there is no such closed-form solution. Current methods typically require the algebraic form of the loss function, its first- and second-order derivatives, and the variance for the estimator, as well as a newly implemented optimization algorithm to obtain the estimator. We use the logistic regression problem as an example. Here, $\boldsymbol{\theta}^* = \arg\min_{\boldsymbol{\theta}} \mathbb{E}[-Y(\boldsymbol{\theta}\mathbf{X})^{\mathrm{T}} - \psi(\mathbf{X}\boldsymbol{\theta})]$ and $\psi(t) = 1/(1 + \exp(-t))$. The ML-assisted estimator is $\widehat{\theta}_{\mathtt{MLA}} = \arg\min_{\boldsymbol{\theta}} \frac{1}{N}\sum_{i=n+1}^{n+N}\widehat{\omega}[-\widehat{f}_i\boldsymbol{\theta}^{\mathrm{T}}\mathbf{X}_i^{\mathrm{T}} - \psi(\mathbf{X}_i\boldsymbol{\theta})] - \{\frac{1}{n}\sum_{i=1}^{n}\widehat{\omega}[-\widehat{f}_i\boldsymbol{\theta}^{\mathrm{T}}\mathbf{X}_i^{\mathrm{T}} - \psi(\mathbf{X}_i\boldsymbol{\theta})] - \frac{1}{n}\sum_{i=1}^{n}[-\widehat{Y}_i\boldsymbol{\theta}^{\mathrm{T}}\mathbf{X}_i^{\mathrm{T}} - \psi(\mathbf{X}_i\boldsymbol{\theta})]\}$ with estimated asymptotic variance $\widehat{\mathbf{A}}^{-1}\widehat{\mathbf{V}}(\omega)\widehat{\mathbf{A}}^{-1}$, where $\widehat{\mathbf{A}} = \frac{1}{N+n}(\sum_{i=1}^{n}\psi''(\mathbf{X}_i\boldsymbol{\theta})\mathbf{X}_i^{\mathrm{T}}\mathbf{X}_i + \sum_{i=n+1}^{n+N}\psi''(\mathbf{X}_i\boldsymbol{\theta})\mathbf{X}_i^{\mathrm{T}}\mathbf{X}_i)$, $\widehat{\mathbf{V}}(\omega) = \frac{n}{N}[\widehat{\omega}^2 \operatorname{Var}_n\left((\psi'(\mathbf{X}_i\boldsymbol{\theta}) - \widehat{f}_i)\mathbf{X}_i^{\mathrm{T}}\right) + \widehat{\operatorname{Cov}}_{N+n}\left((1 - \widehat{\omega})\psi'(\mathbf{X}_i\boldsymbol{\theta})\mathbf{X}_i^{\mathrm{T}} + (\widehat{\omega}\widehat{f}_i - Y_i)\mathbf{X}_i^{\mathrm{T}}\right)$, and $\widehat{\omega}$ needs to be obtained by optimization to minimize the asymptotic variance. In contrast, our protocol simply applies logistic regression $\mathcal{A}$ to

- labeled data with observed outcomes $(\mathbf{X}_{\mathcal{L}}, Y_{\mathcal{L}})$ to obtain $[\widehat{\boldsymbol{\theta}}_{\mathcal{L}}, \widehat{\operatorname{Var}}(\widehat{\boldsymbol{\theta}}_{\mathcal{L}})]$

- labeled data with predicted outcomes $(\mathbf{X}_{\mathcal{L}}, \widehat{f}_{\mathcal{L}})$ to obtain $[\widehat{\boldsymbol{\eta}}_{\mathcal{L}}, \widehat{\operatorname{Var}}(\widehat{\boldsymbol{\eta}}_{\mathcal{L}})]$

- unlabeled data with predicted outcomes $(\mathbf{X}_{\mathcal{U}}, \widehat{f}_{\mathcal{U}})$ to obtain $[\widehat{\boldsymbol{\eta}}_{\mathcal{U}}, \widehat{\operatorname{Var}}(\widehat{\boldsymbol{\eta}}_{\mathcal{U}})]$

- bootstrap of labeled data $(\mathbf{X}_{\mathcal{L}}, Y_{\mathcal{L}}, \widehat{f}_{\mathcal{L}})_q, q = 1, \ldots, Q$ for $\widehat{\operatorname{Cov}}(\widehat{\boldsymbol{\theta}}_{\mathcal{L}}, \widehat{\boldsymbol{\eta}}_{\mathcal{L}})$,

and returns $\widehat{\boldsymbol{\omega}}_0^{\mathrm{T}}\widehat{\boldsymbol{\eta}}_{\mathcal{U}} - (\boldsymbol{\omega}_0^{\mathrm{T}}\widehat{\boldsymbol{\eta}}_{\mathcal{L}} - \widehat{\boldsymbol{\theta}}_{\mathcal{L}})$, where $\widehat{\boldsymbol{\omega}}_0 = (\widehat{\operatorname{Var}}(\widehat{\boldsymbol{\eta}}_{\mathcal{L}}) + \widehat{\operatorname{Var}}(\widehat{\boldsymbol{\eta}}_{\mathcal{U}}))^{-1}\widehat{\operatorname{Cov}}(\widehat{\boldsymbol{\theta}}_{\mathcal{L}}, \widehat{\boldsymbol{\eta}}_{\mathcal{L}})$. For each new statistical task, as long as an existing analysis routine can produce an estimator that is asymptotically normally distributed, our protocol can be similarly applied. Additionally, the current mathematical principles guiding ML-assisted inference apply solely to M-estimation [3, 4, 20, 35, 56]. Our protocol extends beyond this limitation, addressing all estimation problems with an asymptotically normally distributed estimator.

Inference relying solely on summary statistics is widely used in the statistical genetics literature for practical reasons. Summary statistics-based methods have been developed for tasks such as variance component inference and genetic risk prediction [11, 12, 34, 39, 53]. In contrast to our work, these applications do not leverage ML predictions, but instead focus on inference using summary statistics obtained from observed outcomes. An exception is a previous study for valid genome-wide association studies (GWAS) on ML-predicted outcome [36]. However, it focused only on linear regression modeling with application to GWAS. The PSPS framework introduced in this paper aims to extend ML-assisted inference to general statistical tasks.

Our work is also related to semi-supervised learning, resampling-based inference, zero augmentation, and false discovery rate (FDR) control methods. Our protocol is designed for estimation and statistical inference using both labeled and unlabeled data, addressing a different problem from semi-supervised learning [55], which primarily focuses on prediction. Our protocol is inspired by the core principle of resampling-based inference, which replaces algebraic derivation with computation [19]. The main difference is that we focus on how to use ML to support inference, whereas resampling-based inference focuses on bias and variance estimation, and type-I error control. The idea of zero augmentation has been used in augmented inverse propensity weighting estimators [38] and in handling unmeasured confounders [48] and missing data for U-statistics [14]. These estimators do not incorporate ML, which is fundamental to our work. We also adapt techniques from the FDR literature [7–10]. Our unique contribution is to use ML to support FDR control, thereby increasing its statistical power, in contrast to classical methods that rely solely on labeled data.

# 3 Methods

## 3.1 General protocol for PSPS

Building on Section 2, we formalized our protocol in Fig. 1 for ML-assisted inference:

---

**Algorithm 1** PSPS for ML-assisted inference

---

**Input:** A pre-trained ML model $\widehat{f}$, labeled data $\mathcal{L} = (\mathbf{X}_\mathcal{L}, Y_\mathcal{L})$, unlabeled data $\mathcal{U} = \mathbf{X}_\mathcal{U}$

1: Use the ML model $\widehat{f}$ to predict the outcome in both labeled and unlabeled data.
2: Apply the algorithm $\mathcal{A}$ in the analysis routine to

- labeled data $(\mathbf{X}_\mathcal{L}, Y_\mathcal{L})$ and obtain $[\widehat{\boldsymbol{\theta}}_\mathcal{L}, \widehat{\mathrm{Var}}(\widehat{\boldsymbol{\theta}}_\mathcal{L})]$
- labeled data $(\mathbf{X}_\mathcal{L}, \widehat{f}_\mathcal{L})$ and obtain $[\widehat{\boldsymbol{\eta}}_\mathcal{L}, \widehat{\mathrm{Var}}(\widehat{\boldsymbol{\eta}}_\mathcal{L})]$
- unlabeled data with $(\mathbf{X}_\mathcal{U}, \widehat{f}_\mathcal{U})$ and obtain $[\widehat{\boldsymbol{\eta}}_\mathcal{U}, \widehat{\mathrm{Var}}(\widehat{\boldsymbol{\eta}}_\mathcal{U})]$
- $Q$ bootstrap of labeled data $(\mathbf{X}_\mathcal{L}, Y_\mathcal{L}, \widehat{f}_\mathcal{L})_q, q = 1, \ldots, Q$ and obtain $\widehat{\mathrm{Cov}}(\widehat{\boldsymbol{\theta}}_\mathcal{L}, \widehat{\boldsymbol{\eta}}_\mathcal{L})$.

3: Employ one-step debiasing to the summary statistics in step2:
$$\widehat{\boldsymbol{\theta}}_{\mathrm{PSPS}} = \widehat{\boldsymbol{\omega}}_0^{\mathrm{T}} \widehat{\boldsymbol{\eta}}_\mathcal{U} - (\widehat{\boldsymbol{\omega}}_0^{\mathrm{T}} \widehat{\boldsymbol{\eta}}_\mathcal{L} - \widehat{\boldsymbol{\theta}}_\mathcal{L}),$$
where $\widehat{\boldsymbol{\omega}}_0 = [\widehat{\mathrm{Var}}(\widehat{\boldsymbol{\eta}}_\mathcal{L}) + \widehat{\mathrm{Var}}(\widehat{\boldsymbol{\eta}}_\mathcal{U})]^{-1}\widehat{\mathrm{Cov}}(\widehat{\boldsymbol{\theta}}_\mathcal{L}, \widehat{\boldsymbol{\eta}}_\mathcal{L})$ and $\widehat{\mathrm{Var}}(\widehat{\boldsymbol{\theta}}_{\mathrm{PSPS}}) = \widehat{\mathrm{Var}}(\widehat{\boldsymbol{\theta}}_\mathcal{L}) - \widehat{\mathrm{Cov}}(\widehat{\boldsymbol{\theta}}_\mathcal{L}, \widehat{\boldsymbol{\eta}}_\mathcal{L})^{\mathrm{T}}[\widehat{\mathrm{Var}}(\widehat{\boldsymbol{\eta}}_\mathcal{L}) + \widehat{\mathrm{Var}}(\widehat{\boldsymbol{\eta}}_\mathcal{U})]^{-1}\widehat{\mathrm{Cov}}(\widehat{\boldsymbol{\theta}}_\mathcal{L}, \widehat{\boldsymbol{\eta}}_\mathcal{L})$

**Output:** ML-assisted point estimator $\widehat{\boldsymbol{\theta}}_{\mathrm{PSPS}}$, standard error $\sqrt{\widehat{\mathrm{Var}}(\widehat{\boldsymbol{\theta}}_{\mathrm{PSPS}})}$, $\alpha$-level confidence interval for the $k$-th coordinate $\mathcal{C}_{\alpha,k}^{\mathrm{PSPS}} = (\widehat{\boldsymbol{\theta}}_{\mathrm{PSPS}_k} \pm z_{1-\alpha/2}\sqrt{\widehat{\mathrm{Var}}(\widehat{\boldsymbol{\theta}}_{\mathrm{PSPS}})_{kk}})$, and (two-sided) p-value $2(1 - \Phi(|\frac{\widehat{\boldsymbol{\theta}}_{\mathrm{PSPS}_k}}{\sqrt{\widehat{\mathrm{Var}}(\widehat{\boldsymbol{\theta}}_{\mathrm{PSPS}})_{kk}}}|))$, where $\Phi$ is the CDF of the standard normal distribution.

---

The only requirements for our protocol are: i) algorithm $\mathcal{A}$, when applied to labeled data $(\boldsymbol{X}_\mathcal{L}, Y_\mathcal{L})$, returns a consistent and asymptotically normally distributed estimator of $\boldsymbol{\theta}^*$; ii) labeled and unlabeled data are independent and identically distributed. Under these assumptions, the summary statistics have the following asymptotic properties:

$$n^{1/2} \begin{pmatrix} \widehat{\boldsymbol{\theta}}_\mathcal{L} - \boldsymbol{\theta}^* \\ \widehat{\boldsymbol{\eta}}_\mathcal{L} - \boldsymbol{\eta} \\ \widehat{\boldsymbol{\eta}}_\mathcal{U} - \boldsymbol{\eta} \end{pmatrix} \xrightarrow{D} \mathcal{N} \left\{ \begin{pmatrix} \mathbf{0}_K \\ \mathbf{0}_K \\ \mathbf{0}_K \end{pmatrix}, \begin{pmatrix} \mathbf{V}(\widehat{\boldsymbol{\theta}}_\mathcal{L}) & \mathbf{V}(\widehat{\boldsymbol{\theta}}_\mathcal{L}, \widehat{\boldsymbol{\eta}}_\mathcal{L}) & \mathbf{0} \\ \mathbf{V}(\widehat{\boldsymbol{\theta}}_\mathcal{L}, \widehat{\boldsymbol{\eta}}_\mathcal{L}) & \mathbf{V}(\widehat{\boldsymbol{\eta}}_\mathcal{L}) & \mathbf{0} \\ \mathbf{0} & \mathbf{0} & \rho\mathbf{V}(\widehat{\boldsymbol{\eta}}_\mathcal{U}) \end{pmatrix} \right\}, \quad (1)$$

where $\boldsymbol{\eta} \equiv \boldsymbol{\eta}(\mathbb{P}_{\widehat{f}}) \in \mathbb{R}^K$ is defined on $(\mathbf{X}, \widehat{f}) \sim \mathbb{P}_{\widehat{f}}$, $\mathbf{V}(\cdot)$ denotes the asymptotic variance and covariance of a estimator, and $\rho = \frac{n}{N}$. The asymptotic approximation gives $\mathbf{V}(\widehat{\boldsymbol{\theta}}_\mathcal{L}) \approx n \mathrm{Var}(\widehat{\boldsymbol{\theta}}_\mathcal{L}), \mathbf{V}(\widehat{\boldsymbol{\theta}}_\mathcal{L}, \widehat{\boldsymbol{\eta}}_\mathcal{L}) \approx n \mathrm{Cov}(\widehat{\boldsymbol{\theta}}_\mathcal{L}, \widehat{\boldsymbol{\eta}}_\mathcal{L}), \mathbf{V}(\widehat{\boldsymbol{\eta}}_\mathcal{L}) \approx n \mathrm{Var}(\widehat{\boldsymbol{\eta}}_\mathcal{L})$ and $\mathbf{V}(\widehat{\boldsymbol{\eta}}_\mathcal{U}) \approx N \mathrm{Var}(\widehat{\boldsymbol{\eta}}_\mathcal{U})$. Here, we do not require $\widehat{\boldsymbol{\eta}}_\mathcal{L}$ and $\widehat{\boldsymbol{\eta}}_\mathcal{U}$ to be consistent for $\boldsymbol{\theta}^*$, thus allows arbitrary ML model.

With the summary statistics following a multivariate normal distribution asymptotically, the debiased estimator $\widehat{\boldsymbol{\theta}}_{\mathrm{PSPS}} = \widehat{\boldsymbol{\omega}}_0^{\mathrm{T}} \widehat{\boldsymbol{\eta}}_\mathcal{U} - (\widehat{\boldsymbol{\omega}}_0^{\mathrm{T}} \widehat{\boldsymbol{\eta}}_\mathcal{L} - \widehat{\boldsymbol{\theta}}_\mathcal{L})$ is consistent for $\theta^*$ and asymptotically normally distributed (Theorem 1). Therefore, by plugging in a consistent estimator for its asymptotic variance $\mathbf{V}(\widehat{\boldsymbol{\theta}}_{\mathrm{PSPS}}) \approx n \mathrm{Var}(\widehat{\boldsymbol{\theta}}_{\mathrm{PSPS}})$, valid confidence interval and hypothesis testing can be achieved.

*Remark* 1. PSPS is more "task-agnostic" than existing methods in three aspects:

1. For M-estimation tasks, currently, only mean and quantile estimation, as well as linear, logistic, and Poisson regression, have been implemented in software tools and are ready for immediate application. For other M-estimation tasks, task-specific derivation of the ML-assisted loss functions and asymptotic variance via the central limit theorem are necessary. After that, researchers still need to develop software packages and optimization algorithms to carry out real applications. In contrast, PSPS only requires already implemented algorithms and software designed for classical inference based on labeled data.

2. For problems that do not fall under M-estimation but have asymptotically normally distributed estimators, only PSPS can be applied, and all current methods would fail. The principles behind ML-assisted M-estimation do not extend to these tasks.

3. Even for M-estimation tasks that have already been implemented, PSPS offers the additional advantage of relying solely on summary statistics. The "task-specific derivations" refer not only to statistical tasks but also to scientific tasks. Real-world data analysis in any scientific discipline often involves conventions and nuisances that require careful consideration. For example, our work is partly motivated by GWAS [43]. Statistically, GWAS is a linear regression that regresses an outcome on many genetic variants. While the regression-based statistical foundation is simple, conducting a valid GWAS requires accounting for numerous technical issues, such as sample relatedness (i.e., study participants may be genetically related) and population structure (i.e., unrelated individuals of the same ancestry are both genetically and phenotypically similar, creating confounded associations in GWAS). Sophisticated algorithms and software have been developed to address these complex issues [31]. It will be very challenging if all these important features need to be reimplemented in an ML-assisted GWAS framework. With our PSPS protocol, researchers can utilize existing tools that are highly optimized for genetic applications to perform ML-assisted GWAS. This adaptability is not just limited to GWAS, but is a major feature of our approach across scientific domains. PSPS enables researchers to conduct ML-assisted inference using well-established data analysis routines.

*Remark* 2. The "federated data analysis" feature of PSPS refers to the fact that we only require summary statistics as input for inference, rather than individual-level raw data (features $\mathbf{X}$ and label $Y$). For example, consider a scenario where labeled data is in one center and unlabeled data is in another, yet researchers cannot access individual-level data from both centers simultaneously. Under such conditions, current ML-assisted inference, which relies on accessing both labeled and unlabeled data to minimize a joint loss function, is not feasible. However, PSPS circumvents this issue by aggregating summary statistics from multiple centers, thereby performing statistical inference while upholding the privacy of individual-level data.

## 3.2 Theoretical guarantees

In this section, we examine the theoretical properties of PSPS. In what follows, $\xrightarrow{P}$ denotes convergence in probability and $\xrightarrow{D}$ denotes convergence in distribution. All proofs are deferred to the Appendix A.

The first result shows that our proposed estimator is consistent, asymptotically normally distributed, and uniformly better in terms of element-wise asymptotic variance compared with the classical estimator based on labeled data only.

**Theorem 1.** *Assuming equation* (1) *holds, then* $\widehat{\boldsymbol{\theta}}_{\text{PSPS}} \xrightarrow{P} \boldsymbol{\theta}^*$, *and*

$$n^{1/2}(\widehat{\boldsymbol{\theta}}_{\text{PSPS}} - \boldsymbol{\theta}^*) \xrightarrow{D} \mathcal{N}\left(\mathbf{0}, \mathbf{V}(\widehat{\boldsymbol{\theta}}_{\text{PSPS}})\right),$$

*where* $\mathbf{V}(\widehat{\boldsymbol{\theta}}_{\text{PSPS}}) = \mathbf{V}(\widehat{\boldsymbol{\theta}}_{\mathcal{L}}) - \mathbf{V}(\widehat{\boldsymbol{\theta}}_{\mathcal{L}}, \widehat{\boldsymbol{\eta}}_{\mathcal{L}})^{\text{T}}(\mathbf{V}(\widehat{\boldsymbol{\eta}}_{\mathcal{L}}) + \rho \mathbf{V}(\widehat{\boldsymbol{\eta}}_{\mathcal{U}}))^{-1}\mathbf{V}(\widehat{\boldsymbol{\theta}}_{\mathcal{L}}, \widehat{\boldsymbol{\eta}}_{\mathcal{L}})$. *Assume the k-th column of* $\mathbf{V}(\widehat{\boldsymbol{\theta}}_{\mathcal{L}}, \widehat{\boldsymbol{\eta}}_{\mathcal{L}})$ *is not a zero vector and at least one of* $\mathbf{V}(\widehat{\boldsymbol{\eta}}_{\mathcal{L}})$ *and* $\mathbf{V}(\widehat{\boldsymbol{\eta}}_{\mathcal{U}})$ *are positive definite, then* $\mathbf{V}(\widehat{\boldsymbol{\theta}}_{\text{PSPS}})_{kk} \leq \mathbf{V}(\widehat{\boldsymbol{\theta}}_{\mathcal{L}})_{kk}$. *With* $\widehat{\mathbf{V}}(\widehat{\boldsymbol{\theta}}_{\text{PSPS}}) \xrightarrow{P} \mathbf{V}(\widehat{\boldsymbol{\theta}}_{\text{PSPS}})$, $\lim_n \mathbb{P}(\theta_k^* \in \mathcal{C}_{\alpha,k}^{\text{PSPS}}) = 1 - \alpha$.

$\widehat{\mathbf{V}}(\widehat{\boldsymbol{\theta}}_{\text{PSPS}})$ can be obtained by applying the algebraic form of $\mathbf{V}(\widehat{\boldsymbol{\theta}}_{\text{PSPS}})$ using the bootstrap estimators for $\mathbf{V}(\widehat{\boldsymbol{\theta}}_{\mathcal{L}}), \mathbf{V}(\widehat{\boldsymbol{\eta}}_{\mathcal{L}}), \mathbf{V}(\widehat{\boldsymbol{\theta}}_{\mathcal{L}}, \widehat{\boldsymbol{\eta}}_{\mathcal{L}})$, and $\mathbf{V}(\widehat{\boldsymbol{\eta}}_{\mathcal{U}})$. The regularity conditions for consistent bootstrap variance estimation are outlined in Theorem 3.10 (i) of [41]. We also refer readers to [21], which showed that bootstrap-based variance provides valid but potentially conservative inference.

This result indicates that a greater reduction in variance for the ML-assisted estimator is associated with larger values of $\mathbf{V}(\widehat{\boldsymbol{\theta}}_{\mathcal{L}}, \widehat{\boldsymbol{\eta}}_{\mathcal{L}})$ and smaller values of $\mathbf{V}(\widehat{\boldsymbol{\eta}}_{\mathcal{L}}), \mathbf{V}(\widehat{\boldsymbol{\eta}}_{\mathcal{U}})$, and $\rho$. The variance reduction term $[\mathbf{V}(\widehat{\boldsymbol{\theta}}_{\mathcal{L}}, \widehat{\boldsymbol{\eta}}_{\mathcal{L}})^{\text{T}}(\mathbf{V}(\widehat{\boldsymbol{\eta}}_{\mathcal{L}}) + \rho \mathbf{V}(\widehat{\boldsymbol{\eta}}_{\mathcal{U}}))^{-1}\mathbf{V}(\widehat{\boldsymbol{\theta}}_{\mathcal{L}}, \widehat{\boldsymbol{\eta}}_{\mathcal{L}})]_{kk}$ can also serve as a metric for selecting the optimal ML model in ML-assisted inference.

Our next result shows that three existing methods, i.e., PPI, PPI++, and PSPA, are asymptotically equivalent to PSPS with different weighting matrices. A broader class for consistent estimator of $\boldsymbol{\theta}^*$ is $\widehat{\boldsymbol{\theta}}(\boldsymbol{\omega}) = \boldsymbol{\omega}^{\mathrm{T}}\widehat{\boldsymbol{\eta}}_{\mathcal{U}} - (\boldsymbol{\omega}^{\mathrm{T}}\widehat{\boldsymbol{\theta}}_{\mathcal{L}} - \widehat{\boldsymbol{\eta}}_{\mathcal{L}})$, where $\boldsymbol{\omega}$ is a $K \times K$ matrix. The consistency of $\widehat{\boldsymbol{\theta}}(\boldsymbol{\omega})$ for $\boldsymbol{\theta}^*$ only requires $\boldsymbol{\omega}^{\mathrm{T}}(\widehat{\boldsymbol{\eta}}_{\mathcal{U}} - \widehat{\boldsymbol{\eta}}_{\mathcal{L}}) \xrightarrow{P} \mathbf{0}$. Since $(\widehat{\boldsymbol{\eta}}_{\mathcal{U}} - \widehat{\boldsymbol{\eta}}_{\mathcal{L}}) \xrightarrow{P} \mathbf{0}$, assigning arbitrarily fixed weights for will satisfy the condition. However, the choice of weights influences the efficiency of the estimator as illustrated in Proposition 2 later.

**Proposition 1.** *Assuming equation* (1) *and regularity condition for the asymptotic normality of current ML-assisted estimator holds. For any M-estimation problem, we have*

$$n^{\frac{1}{2}}(\widehat{\boldsymbol{\theta}}(\operatorname{diag}(\boldsymbol{\omega}_{\mathrm{ele}})\mathbf{C}) - \widehat{\boldsymbol{\theta}}_{\mathtt{PSPA}}) \xrightarrow{D} \mathbf{0}, n^{\frac{1}{2}}(\widehat{\boldsymbol{\theta}}(\operatorname{diag}(\boldsymbol{\omega}_{\mathrm{tr}})\mathbf{C}) - \widehat{\boldsymbol{\theta}}_{\mathtt{PPI++}}) \xrightarrow{D} \mathbf{0}, n^{\frac{1}{2}}(\widehat{\boldsymbol{\theta}}(\operatorname{diag}(\mathbf{1})\mathbf{C}) - \widehat{\boldsymbol{\theta}}_{\mathtt{PPI}}) \xrightarrow{D} \mathbf{0}.$$

*Here, $\boldsymbol{\omega}_{\mathrm{ele}} = [\omega_{\mathrm{ele},1}, \ldots, \omega_{\mathrm{ele},K}]^{\mathrm{T}} \in \mathbb{R}^K$ and $\omega_{\mathrm{ele},k}$ minimizing the k-th diagonal element of $\mathbf{V}(\widehat{\boldsymbol{\theta}}(\boldsymbol{\omega}))$, $\omega_{tr}$ is a scalar used to minimize the trace of $\mathbf{V}(\widehat{\boldsymbol{\theta}}(\boldsymbol{\omega}))$, and $\mathbf{C}$ is a matrix associated with the second derivatives of the loss function in M-estimation, with further details deferred to Appendix A.*

This demonstrates that for M-estimation problems, our method is asymptotically equivalent to PSPA, PPI++, and PPI with the respective weights $\operatorname{diag}(\boldsymbol{\omega}_{\mathrm{ele}})\mathbf{C}$, $\operatorname{diag}(\boldsymbol{\omega}_{\mathrm{tr}})\mathbf{C}$, and $\operatorname{diag}(\mathbf{1})\mathbf{C}$. Therefore, PSPS can be viewed as a generalization of these existing methods.

Our third result shows that the weights used in the Proposition 1 are not optimal. Instead, our choice of $\boldsymbol{\omega}_0$ represents the optimal smooth combination of $(\widehat{\boldsymbol{\theta}}_{\mathcal{L}}, \widehat{\boldsymbol{\eta}}_{\mathcal{L}}, \widehat{\boldsymbol{\eta}}_{\mathcal{U}})$ in terms of minimizing the asymptotic variance, while still preserving consistency.

**Proposition 2.** *Suppose $n^{1/2}(g(\widehat{\boldsymbol{\theta}}_{\mathcal{L}}, \widehat{\boldsymbol{\eta}}_{\mathcal{L}}, \widehat{\boldsymbol{\eta}}_{\mathcal{U}}) - \boldsymbol{\theta}^*) \xrightarrow{D} \mathcal{N}(0, \boldsymbol{\Sigma}_g)$ and g is a smooth function, then $\boldsymbol{\Sigma}_{g_{kk}} \geq \boldsymbol{\Sigma}_{\mathtt{PSPS}_{kk}}$*

Together with Proposition 1, our results demonstrate that our protocol provides a more efficient estimator compared to existing methods for the M-estimation problems. Furthermore, the applicability of our protocol is not limited to M-estimation and only requires summary statistics as input. It also indicates that in a setting of federated data analysis [24] where individual-level data are not available, PSPS proves to be the optimal approach for combining shared summary statistics.

*Remark* 3. PPI++ [4] employs a power-tuning scalar for variance reduction in ML-assisted inference. This scalar is obtained by minimizing the trace or possibly other scalarization of the estimator's variance-covariance matrix. However, the asymptotic variance of PSPS is always equal to or smaller than that of PPI++, irrespective of the scalarization chosen by researchers. This advantage arises because PSPS utilizes a $K \times K$ power tuning matrix, $\boldsymbol{\omega}$, for variance reduction, where $K$ represents the dimensionality of parameters. This matrix facilitates information sharing across different parameter coordinates, thereby enhancing estimation precision. The choice of weighting matrix in PSPS also allows for element-wise variance reduction, reducing each diagonal element of the variance-covariance matrix. In contrast, the single scalar in PPI++ can only target overall trace reduction or variance reduction of a specific element. A detailed example is provided in Appendix B. Only in one-dimensional parameter estimation tasks, such as mean estimation, PPI++ and PSPS exhibit the same asymptotic variance.

### 3.3 Extensions

We also provide several extensions to ensure the broad applicability of our method.

#### 3.3.1 Labeled data and unlabeled data are not independent

Here, we relax the assumption that the labeled data and unlabeled data are independent. When they are not independent, this can lead to the non-zero covariance between the $\widehat{\boldsymbol{\eta}}_{\mathcal{L}}$ and $\widehat{\boldsymbol{\eta}}_{\mathcal{U}}$. Consider a broader class of summary statistics asymptotically satisfying

$$n^{1/2}\begin{pmatrix} \widehat{\boldsymbol{\theta}}_{\mathcal{L}} - \boldsymbol{\theta}^* \\ \widehat{\boldsymbol{\eta}}_{\mathcal{L}} - \boldsymbol{\eta} \\ \widehat{\boldsymbol{\eta}}_{\mathcal{U}} - \boldsymbol{\eta} \end{pmatrix} \xrightarrow{D} \mathcal{N}\left\{\begin{pmatrix} \mathbf{0}_K \\ \mathbf{0}_K \\ \mathbf{0}_K \end{pmatrix}, \left(\begin{pmatrix} \mathbf{V}(\widehat{\boldsymbol{\theta}}_{\mathcal{L}}) & \mathbf{V}(\widehat{\boldsymbol{\theta}}_{\mathcal{L}}, \widehat{\boldsymbol{\eta}}_{\mathcal{L}}) & \mathbf{V}(\widehat{\boldsymbol{\eta}}_{\mathcal{L}}, \widehat{\boldsymbol{\eta}}_{\mathcal{U}}) \\ \mathbf{V}(\widehat{\boldsymbol{\theta}}_{\mathcal{L}}, \widehat{\boldsymbol{\eta}}_{\mathcal{L}}) & \mathbf{V}(\widehat{\boldsymbol{\eta}}_{\mathcal{L}}) & \sqrt{\rho}\mathbf{V}(\widehat{\boldsymbol{\theta}}_{\mathcal{L}}, \widehat{\boldsymbol{\eta}}_{\mathcal{U}}) \\ \mathbf{V}(\widehat{\boldsymbol{\eta}}_{\mathcal{L}}, \widehat{\boldsymbol{\eta}}_{\mathcal{U}}) & \sqrt{\rho}\mathbf{V}(\widehat{\boldsymbol{\theta}}_{\mathcal{L}}, \widehat{\boldsymbol{\eta}}_{\mathcal{U}}) & \rho\mathbf{V}(\widehat{\boldsymbol{\eta}}_{\mathcal{U}}) \end{pmatrix}\right)\right\}$$

We can similarly employ the one-step debiasing $\widehat{\boldsymbol{\theta}}_{\texttt{PSPS}}^{\texttt{no-indep}} = \widehat{\boldsymbol{\omega}}_0^{\mathrm{T}} \widehat{\boldsymbol{\eta}}_{\mathcal{U}} - \widehat{\boldsymbol{\omega}}_0(\widehat{\boldsymbol{\eta}}_{\mathcal{L}} - \widehat{\boldsymbol{\theta}})$ where $\widehat{\boldsymbol{\omega}}_0 = (\widehat{\mathbf{V}}(\widehat{\boldsymbol{\theta}}_{\mathcal{L}}, \widehat{\boldsymbol{\eta}}_{\mathcal{L}}) - \widehat{\mathbf{V}}(\widehat{\boldsymbol{\eta}}_{\mathcal{L}}, \widehat{\boldsymbol{\eta}}_{\mathcal{U}}))^{\mathrm{T}}(\widehat{\mathbf{V}}(\widehat{\boldsymbol{\eta}}_{\mathcal{L}}) + \widehat{\mathbf{V}}(\widehat{\boldsymbol{\eta}}_{\mathcal{U}}) - 2\widehat{\mathbf{V}}(\widehat{\boldsymbol{\theta}}_{\mathcal{L}}, \widehat{\boldsymbol{\eta}}_{\mathcal{U}}))^{-1}$ and $\widehat{\mathrm{Var}}(\widehat{\boldsymbol{\theta}}_{\texttt{PSPS}}^{\texttt{no-indep}}) = \widehat{\mathrm{Var}}(\widehat{\boldsymbol{\theta}}_{\mathcal{L}}) - (\widehat{\mathbf{V}}(\widehat{\boldsymbol{\theta}}_{\mathcal{L}}, \widehat{\boldsymbol{\eta}}_{\mathcal{L}}) - \widehat{\mathbf{V}}(\widehat{\boldsymbol{\eta}}_{\mathcal{L}}, \widehat{\boldsymbol{\eta}}_{\mathcal{U}}))^{\mathrm{T}}(\mathbf{V}(\widehat{\boldsymbol{\eta}}_{\mathcal{L}}) + \widehat{\mathbf{V}}(\widehat{\boldsymbol{\eta}}_{\mathcal{U}}) - 2\widehat{\mathbf{V}}(\widehat{\boldsymbol{\theta}}_{\mathcal{L}}, \widehat{\boldsymbol{\eta}}_{\mathcal{U}}))^{-1}(\widehat{\mathbf{V}}(\widehat{\boldsymbol{\theta}}_{\mathcal{L}}, \widehat{\boldsymbol{\eta}}_{\mathcal{L}}) - \widehat{\mathbf{V}}(\widehat{\boldsymbol{\eta}}_{\mathcal{L}}, \widehat{\boldsymbol{\eta}}_{\mathcal{U}}))$. The theoretical guarantees of the proposed estimator can be similarly derived by Theorem 1.

### 3.3.2 Sensitivity analysis for distributional shift between labeled and unlabeled data

The other assumption of our approach is that the labeled and unlabeled data are identically distributed so that we can ensure $\widehat{\boldsymbol{\eta}}_{\mathcal{L}} - \widehat{\boldsymbol{\eta}}_{\mathcal{U}} \xrightarrow{P} \mathbf{0}$ and validity of PSPS results. To address the potential violation of this assumption, we introduce a sensitivity analysis with hypothesis testing for the null $H_0 : \boldsymbol{\eta}_{\mathcal{L},k} = \boldsymbol{\eta}_{\mathcal{U},k}$ with test statistics $\frac{\widehat{\eta}_{\mathcal{L},k} - \widehat{\eta}_{\mathcal{U},k}}{\sqrt{\widehat{\mathrm{Var}}(\widehat{\eta}_{\mathcal{L},k}) + \widehat{\mathrm{Var}}(\widehat{\eta}_{\mathcal{U},k})}} \xrightarrow{D} \mathcal{N}(0,1)$ to assess if $\boldsymbol{\eta}_{\mathcal{L},k}$ and $\boldsymbol{\eta}_{\mathcal{U},k}$ are significantly different. Here, the subscript $k$ indicates the $k$-th coordinate. We recommend using p-value $< 0.1$ as evidence for heterogeneity and caution the interpretation of results from ML-assisted inference.

### 3.3.3 ML-assisted FDR control

The output from PSPS can be used for ML-assisted FDR control, achieving greater statistical power compared to classical FDR control methods that solely rely on labeled data. We refer to our approach as PSPS-BH and PSPS-knockoff. Briefly, PSPS-BH processes the p-value from ML-assisted linear regression through the Benjamini-Hochberg (BH) procedure [9], while PSPS-knockoff utilizes the ML-assisted debiased Lasso coefficient [23, 51] in the ranking algorithm of knockoff [7]. We present our algorithm in Appendix C and evaluate their performance using experiments in Section 4.

### 3.3.4 ML-assisted inference with predicted features

We have discussed ML-assisted inference with outcomes predicted by ML models. Here, we note that PSPS can also be applied when either features alone are predicted or both features and outcomes are predicted. The key idea is that the difference between point estimators obtained from applying $\mathcal{A}$ to predicted features in both labeled and unlabeled datasets is a consistent estimator for zero. This enables zero augmentation for estimators from observed features and outcomes. To implement this, modify step 2 in Algorithm 1 to apply $\mathcal{A}$ to predicted features in both labeled and unlabeled data. A similar approach is applicable when both features and outcomes are predicted.

## 4 Numerical experiments and real data application

### 4.1 Simulations

We conduct simulations to assess the finite sample performance of our method. Our objectives are to demonstrate that 1) PSPS achieves narrower confidence intervals when applied to statistical tasks already implemented in existing ML-assisted methods; 2) when applied to statistical tasks that have not been implemented for ML-assisted inference, PSPS provides confidence intervals with narrower width but correct coverage (indicating higher statistical power) compared to classical approaches rely solely on labeled data; 3) PSPS provides well-calibrated FDR control and achieves higher power compared to classical methods only using labeled data.

**Tasks that have been implemented for ML-assisted inference** We compare PSPS with the classical method using only labeled data, the imputation-based method using only unlabeled data, and three ML-assisted inference methods PPI, PPI++, and PSPA [3, 4, 35] on mean estimation and linear and logistic regression. We defer the detailed data-generating process to Appendix D. In short, we generated outcome $Y_i$ from feature $X_{1i}$ and $X_{2i}$, and obtained the pre-trained random forest that predict $Y_i$ using $X_{1i}$ and $X_{2i}$. We have 500 labeled samples $(X_{1i}, Y_i, \widehat{f}_i)$, and unlabeled samples $(X_{1i}, \widehat{f}_i)$ ranged from 1,000 to 10,000. Our goal is to estimate the mean of $Y_i$, as well as the linear and logistic regression coefficient between $Y_i$ and $X_{1i}$.

Fig. 2a-c show confidence interval coverage and Fig. 2d-f show confidence interval width. We find that the imputation-based method fails to obtain correct coverage, while all others including PSPS have the correct coverage. PSPS has narrower confidence intervals compared to the classical method and other approaches for ML-assisted inference.

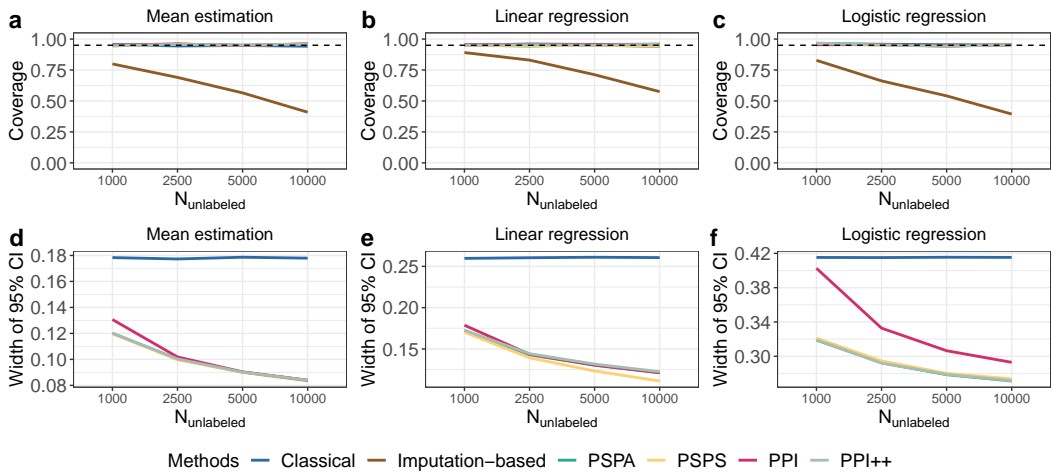

Figure 2: Simulation for tasks that have been implemented for ML-assisted inference including mean estimation, linear regression, and logistic regression from left to right. Panel a-c present confidence interval coverage and panels d-f present confidence interval width.

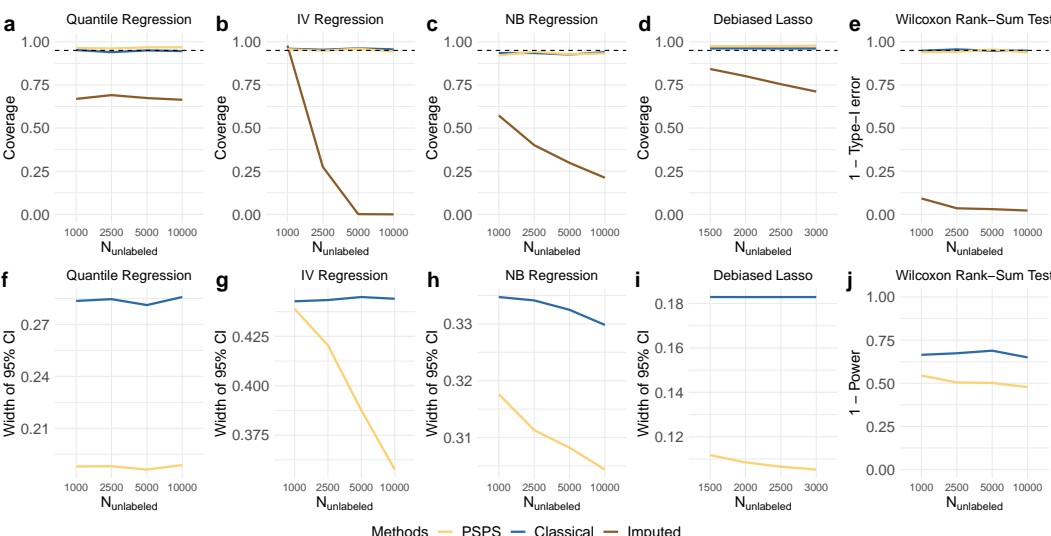

Figure 3: Simulation for tasks that have not been implemented for ML-assisted inference including quantile regression, instrumental variable (IV) regression, negative binomial (NB) regression, debiased Lasso, and Wilcoxon rank-sum test from left to right. Panels a-e present confidence interval coverage (1 - type-error for Wilcoxon rank-sum test) and panels f-j present confidence interval width (1 - power for Wilcoxon rank-sum test).

**Tasks that have not been implemented for ML-assisted inference** Next, we consider several commonly used statistical tasks that currently lack implementations for ML-assisted inference including quantile regression [27], instrumental variable (IV) regression [5], negative binomial (NB) regression [22], debiased Lasso [51], and the Wilcoxon rank-sum test [28]. Similar to our previous simulations, we utilize labeled data, unlabeled data, and a pre-trained ML model. Detailed simulation settings are deferred to the Appendix D. Our goal is to estimate the regression coefficient between $Y_i$ and $X_{1i}$ for quantile (at quantile level 0.75), IV, and NB regression, between $Y_i$ and high dimensional features $\mathbf{X}_i \in \mathbb{R}^{150}$ for debiased Lasso, and to perform hypothesis testing on the medians of two independent samples $Y_i|X_{1i} = 1$ and $Y_i|X_{1i} = 0$ using the Wilcoxon rank-sum test.

Fig. 3a-d show confidence interval coverage and Fig. 3f-i show confidence interval width for parameter estimation. Fig. 3e and Fig. 3j show the type-I error and statistical power for the Wilcoxon rank-sum test. We found that the imputation-based method fails to obtain correct confidence interval coverage and shows inflated type-I error, while PSPS and classical method have the correct coverage and well-calibrated type-I error control. PSPS has narrower confidence intervals width in all settings, and higher statistical power for the Wilcoxon rank-sum test compared to classical methods. Confidence intervals become narrower as unlabeled sample size increases, indicating higher efficiency gain.

**FDR control** We evaluate the finite sample performance of PSPS-BH and PSPS-knockoff in controlling the FDR compared with classical and imputation-based methods as baselines. We consider low–dimensional($K < n$) and high-dimensional($K > n$) linear regressions for PSPS-BH and PSPS-knockoff, respectively. We simulate the data such that only a proportion of the features are truly associated with the outcome. The data generating process is deferred to Appendix D. Our goal is to select the associated features while maintaining the target FDR level.

Fig. E.1a-b shows the estimated FDR and Fig. E.1c-d shows the statistical power for different methods. Imputation-based method failed to control FDR in either low-dimensional or high-dimensional settings. Classical approach, PSPS-BH, and PSPS-knockoff effectively controlled in both low-dimensional and high-dimensional settings. PSPS-BH, and PSPS-knockoff achieve higher statistical power compared to the classical method.

These simulations demonstrate that PSPS outperforms existing methods and can be easily adapted for various statistical tasks not yet implemented in current ML-assisted inference methods.

## 4.2 Identify vQTLs for bone mineral density

We employed our method to carry out ML-assisted quantile regression to identify genetic variants associated with the outcome variability (vQTL) of bone mineral density derived from dual-energy X-ray absorptiometry imaging (DXA-BMD) [33]. DXA-BMD is the primary diagnostic marker for osteoporosis and fracture risk [15, 54]. Identifying vQTL for DXA-BMD can provide insights into the biological mechanisms underlying outcome plasticity and reveal candidate genetic variants involved in potential gene-gene and gene-environment interactions [29, 32, 45, 47, 49]. We focused on total body DXA-BMD, which integrates measurements from multiple skeletal sites. We used data from the UK Biobank [13], which includes 36,971 labeled and 319,548 unlabeled samples with 9,450,880 genetic variants after quality control. We predicted DXA-BMD values in both labeled and unlabeled samples using SoftImpute [30] with 466 other variables measured in the UK Biobank. Prediction in the labeled sample was implemented through cross-validation to avoid overfitting. The implementation detail is deferred to Appendix D. We used the BH procedure to correct for multiple testing and considered FDR $< 0.05$ as the significance threshold.

No genetic variants reached statistical significance under the classical method with only labeled data. PSPS identified 108 significant variants with FDR $< 0.05$ spanning 5 independent loci, showcasing the superior statistical power of PSPS (Fig. E.2 and Table E.1). Notably, these significant vQTL cannot be identified by linear regression [36], indicating the different genetic mechanisms controlling outcome levels and variability for DXA-BMD.

## 4.3 Computational efficiency

We compared the computational efficiency of PSPS with existing methods using a dataset of 500 labeled and 10,000 unlabeled data points. Results are shown in Table E.2. While PSPS is slower due to resampling, its overall runtime is still relatively short.

## 5 Conclusion

We introduced a simple, task-agnostic protocol for ML-assisted inference, with applications across a broad range of statistical tasks. We established the consistency and asymptotic normality of the proposed estimator. We further introduced several extensions to expand the scope of our approach. Through extensive experiments, we demonstrated the superior performance and broad applicability of our method across diverse tasks. Our protocol involves initially generating summary statistics using computationally efficient software tools in scientific data analysis, followed by integration of summary statistics to produce ML-assisted inference results, which achieves high computational efficiency while maintaining statistical validity. Future work could focus on developing a fast resampling algorithm to further improve computational efficiency.

**Acknowledgements:** We gratefully acknowledge research support from the National Institutes of Health (NIH; grant U01 HG012039) and support from the University of Wisconsin–Madison Office of the Chancellor and the Vice Chancellor for Research and Graduate Education with funding from the Wisconsin Alumni Research Foundation.

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

# A Proofs

## A.1 Proof the Theorem 1

*Proof.* Since

$$n^{1/2} \begin{pmatrix} \widehat{\boldsymbol{\theta}}_{\mathcal{L}} - \boldsymbol{\theta}^* \\ \widehat{\boldsymbol{\eta}}_{\mathcal{L}} - \boldsymbol{\eta} \\ \widehat{\boldsymbol{\eta}}_{\mathcal{U}} - \boldsymbol{\eta} \end{pmatrix} \xrightarrow{D} \mathcal{N} \left\{ \begin{pmatrix} \mathbf{0}_K \\ \mathbf{0}_K \\ \mathbf{0}_K \end{pmatrix}, \begin{pmatrix} \mathbf{V}(\widehat{\boldsymbol{\theta}}_{\mathcal{L}}) & \mathbf{V}(\widehat{\boldsymbol{\theta}}_{\mathcal{L}}, \widehat{\boldsymbol{\eta}}_{\mathcal{L}}) & \mathbf{0} \\ \mathbf{V}(\widehat{\boldsymbol{\theta}}_{\mathcal{L}}, \widehat{\boldsymbol{\eta}}_{\mathcal{L}}) & \mathbf{V}(\widehat{\boldsymbol{\eta}}_{\mathcal{L}}) & \mathbf{0} \\ \mathbf{0} & \mathbf{0} & \rho \mathbf{V}(\widehat{\boldsymbol{\eta}}_{\mathcal{U}}) \end{pmatrix} \right\}, \quad (2)$$

$\widehat{\boldsymbol{\theta}}_{\mathcal{L}} \xrightarrow{P} \boldsymbol{\theta}^*$ and $\widehat{\boldsymbol{\eta}}_{\mathcal{U}} - \widehat{\boldsymbol{\eta}}_{\mathcal{L}} \xrightarrow{P} \mathbf{0}$. Given weights $\widehat{\boldsymbol{\omega}}_0 = (\widehat{\mathbf{V}}(\widehat{\boldsymbol{\eta}}_{\mathcal{L}}) + \rho \widehat{\mathbf{V}}(\widehat{\boldsymbol{\eta}}_{\mathcal{U}}))^{-1} \widehat{\mathbf{V}}(\widehat{\boldsymbol{\theta}}_{\mathcal{L}}, \widehat{\boldsymbol{\eta}}_{\mathcal{L}})$, where the variances are consistently estimated, Slutsky's theorem implies that, $\widehat{\boldsymbol{\omega}}_0^{\mathrm{T}} (\widehat{\boldsymbol{\eta}}_{\mathcal{U}} - \widehat{\boldsymbol{\eta}}_{\mathcal{L}}) \xrightarrow{P} \mathbf{0}$.

Also by Slutsky's theorem,

$$\widehat{\boldsymbol{\theta}}_{\mathrm{PSPS}} = \widehat{\boldsymbol{\theta}}_{\mathcal{L}} + \widehat{\boldsymbol{\omega}}_0^{\mathrm{T}} (\widehat{\boldsymbol{\eta}}_{\mathcal{U}} - \widehat{\boldsymbol{\eta}}_{\mathcal{L}}) \xrightarrow{P} \boldsymbol{\theta}^*, \quad (3)$$

which completes the proof of consistency.

By multivariate delta methods, denoting $h([\mathbf{x}, \mathbf{y}, \mathbf{z}]^{\mathrm{T}}) = \mathbf{x} + \boldsymbol{\omega}_0^{\mathrm{T}} (\mathbf{z} - \mathbf{y})$, we have $\nabla h([\mathbf{x}, \mathbf{y}, \mathbf{z}]^{\mathrm{T}}) = [\mathbf{1}, -\boldsymbol{\omega}_0, \boldsymbol{\omega}_0]$, therefore by the consistency of $\widehat{\boldsymbol{\omega}}_0$,

$$n^{1/2} h[\begin{pmatrix} \widehat{\boldsymbol{\theta}}_{\mathcal{L}} - \boldsymbol{\theta}^* \\ \widehat{\boldsymbol{\eta}}_{\mathcal{L}} - \boldsymbol{\eta} \\ \widehat{\boldsymbol{\eta}}_{\mathcal{U}} - \boldsymbol{\eta} \end{pmatrix}] = n^{1/2} [\widehat{\boldsymbol{\theta}}_{\mathcal{L}} + \boldsymbol{\omega}_0^{\mathrm{T}} (\widehat{\boldsymbol{\eta}}_{\mathcal{U}} - \widehat{\boldsymbol{\eta}}_{\mathcal{L}})] \quad (4)$$

$$\xrightarrow{D} \mathcal{N}(\boldsymbol{\theta}^*, \mathbf{V}(\widehat{\boldsymbol{\theta}}_{\mathcal{L}}) - \mathbf{V}(\widehat{\boldsymbol{\theta}}_{\mathcal{L}}, \widehat{\boldsymbol{\eta}}_{\mathcal{L}})^{\mathrm{T}} (\mathbf{V}(\widehat{\boldsymbol{\eta}}_{\mathcal{L}}) + \rho \mathbf{V}(\widehat{\boldsymbol{\eta}}_{\mathcal{U}}))^{-1} \mathbf{V}(\widehat{\boldsymbol{\theta}}_{\mathcal{L}}, \widehat{\boldsymbol{\eta}}_{\mathcal{L}})), \quad (5)$$

which completes the proof of asymptotic normality.

Denote $\mathbf{V}(\widehat{\boldsymbol{\theta}}_{\mathcal{L}}, \widehat{\boldsymbol{\eta}}_{\mathcal{L}})_{:,k}$ as the $k$-th column of $\mathbf{V}(\widehat{\boldsymbol{\theta}}_{\mathcal{L}}, \widehat{\boldsymbol{\eta}}_{\mathcal{L}})$, the $k$-th diagonal element of $\mathbf{V}(\widehat{\boldsymbol{\theta}}_{\mathcal{L}}, \widehat{\boldsymbol{\eta}}_{\mathcal{L}})^{\mathrm{T}} (\mathbf{V}(\widehat{\boldsymbol{\eta}}_{\mathcal{L}}) + \rho \mathbf{V}(\widehat{\boldsymbol{\eta}}_{\mathcal{U}}))^{-1} \mathbf{V}(\widehat{\boldsymbol{\theta}}_{\mathcal{L}}, \widehat{\boldsymbol{\eta}}_{\mathcal{L}})$ is a quadratic form

$$\mathbf{V}(\widehat{\boldsymbol{\theta}}_{\mathcal{L}}, \widehat{\boldsymbol{\eta}}_{\mathcal{L}})_{:,k}^{\mathrm{T}} (\mathbf{V}(\widehat{\boldsymbol{\eta}}_{\mathcal{L}}) + \rho \mathbf{V}(\widehat{\boldsymbol{\eta}}_{\mathcal{U}}))^{-1} \mathbf{V}(\widehat{\boldsymbol{\theta}}_{\mathcal{L}}, \widehat{\boldsymbol{\eta}}_{\mathcal{L}})_{:,k}. \quad (6)$$

Here, by our assumption, $(\mathbf{V}(\widehat{\boldsymbol{\eta}}_{\mathcal{L}}) + \rho \mathbf{V}(\widehat{\boldsymbol{\eta}}_{\mathcal{U}}))^{-1}$ is a positive definite matrix. Therefore, quadratic form $\mathbf{V}(\widehat{\boldsymbol{\theta}}_{\mathcal{L}}, \widehat{\boldsymbol{\eta}}_{\mathcal{L}})_{:,k}^{\mathrm{T}} (\mathbf{V}(\widehat{\boldsymbol{\eta}}_{\mathcal{L}}) + \rho \mathbf{V}(\widehat{\boldsymbol{\eta}}_{\mathcal{U}}))^{-1} \mathbf{V}(\widehat{\boldsymbol{\theta}}_{\mathcal{L}}, \widehat{\boldsymbol{\eta}}_{\mathcal{L}})_{:,k}$ is non-negative and is zero if only all elements of $\mathbf{V}(\widehat{\boldsymbol{\theta}}_{\mathcal{L}}, \widehat{\boldsymbol{\eta}}_{\mathcal{L}})_{:,k}$ is zero, which completes the proof of element-wise variance reduction. $\square$

## A.2 Proof the Proposition 1

*Proof.* Given

$$n^{1/2} \begin{pmatrix} \widehat{\boldsymbol{\theta}}_{\mathcal{L}} - \boldsymbol{\theta}^* \\ \widehat{\boldsymbol{\eta}}_{\mathcal{L}} - \boldsymbol{\eta} \\ \widehat{\boldsymbol{\eta}}_{\mathcal{U}} - \boldsymbol{\eta} \end{pmatrix} \xrightarrow{D} \mathcal{N} \left\{ \begin{pmatrix} \mathbf{0}_K \\ \mathbf{0}_K \\ \mathbf{0}_K \end{pmatrix}, \begin{pmatrix} \mathbf{V}(\widehat{\boldsymbol{\theta}}_{\mathcal{L}}) & \mathbf{V}(\widehat{\boldsymbol{\theta}}_{\mathcal{L}}, \widehat{\boldsymbol{\eta}}_{\mathcal{L}}) & \mathbf{0} \\ \mathbf{V}(\widehat{\boldsymbol{\theta}}_{\mathcal{L}}, \widehat{\boldsymbol{\eta}}_{\mathcal{L}}) & \mathbf{V}(\widehat{\boldsymbol{\eta}}_{\mathcal{L}}) & \mathbf{0} \\ \mathbf{0} & \mathbf{0} & \rho \mathbf{V}(\widehat{\boldsymbol{\eta}}_{\mathcal{U}}) \end{pmatrix} \right\}, \quad (7)$$

the asymptotic variance of $\widehat{\boldsymbol{\theta}}(\boldsymbol{\omega}) = \widehat{\boldsymbol{\theta}}(\boldsymbol{\omega}) = \widehat{\boldsymbol{\theta}}_{\mathcal{L}} + \boldsymbol{\omega}^{\mathrm{T}} (\widehat{\boldsymbol{\eta}}_{\mathcal{U}} - \widehat{\boldsymbol{\eta}}_{\mathcal{L}})$ is

$$\mathbf{V}(\widehat{\boldsymbol{\theta}}(\boldsymbol{\omega})) = \mathbf{V}(\widehat{\boldsymbol{\theta}}_{\mathcal{L}}) + \boldsymbol{\omega}^{\mathrm{T}} \mathbf{V}(\widehat{\boldsymbol{\eta}}_{\mathcal{L}}) \boldsymbol{\omega} + \boldsymbol{\omega}^{\mathrm{T}} \rho \mathbf{V}(\widehat{\boldsymbol{\eta}}_{\mathcal{U}}) \boldsymbol{\omega} - 2 \boldsymbol{\omega}^{\mathrm{T}} \mathbf{V}(\widehat{\boldsymbol{\theta}}_{\mathcal{L}}, \widehat{\boldsymbol{\eta}}_{\mathcal{L}}) \quad (8)$$

We first define the M-estimation (Z-estimation) problem. The goal is to estimate a $K$-dimensional parameter $\boldsymbol{\theta}^*$ defined through an estimating equation

$$\mathbb{E}\{\boldsymbol{\psi}(Y, \mathbf{X}; \boldsymbol{\theta})\} = \mathbf{0}, \quad (9)$$

where $\boldsymbol{\psi}(\cdot, \cdot; \boldsymbol{\theta})$ is a user-defined function. By the theory of Z-estimation and a recent paper on ML-assisted inference [35], we have $\mathbf{V}(\widehat{\boldsymbol{\theta}}_{\mathcal{L}}) = \mathbf{A}_{\boldsymbol{\theta}^*}^{-1} \mathbf{M}_1 \mathbf{A}_{\boldsymbol{\theta}^*}^{-1}, \mathbf{V}(\widehat{\boldsymbol{\theta}}_{\mathcal{L}}, \widehat{\boldsymbol{\eta}}_{\mathcal{L}}) = \mathbf{A}_{\boldsymbol{\eta}}^{-1} \mathbf{M}_4 \mathbf{A}_{\boldsymbol{\theta}^*}^{-1}, \mathbf{V}(\widehat{\boldsymbol{\eta}}_{\mathcal{L}}) = \mathbf{A}_{\boldsymbol{\eta}}^{-1} \mathbf{M}_2 \mathbf{A}_{\boldsymbol{\eta}}^{-1}$, and $\mathbf{V}(\widehat{\boldsymbol{\eta}}_{\mathcal{U}}) = \mathbf{A}_{\boldsymbol{\eta}}^{-1} \mathbf{M}_3 \mathbf{A}_{\boldsymbol{\eta}}^{-1}$. Here, $\mathbf{M}_1 = \mathrm{Var}_l[\boldsymbol{\psi}(Y, \mathbf{X}; \boldsymbol{\theta}^*)], \mathbf{M}_2 =$

$\mathrm{Var}_l[\boldsymbol{\psi}(\widehat{f}, \mathbf{X}; \boldsymbol{\theta}^*)], \mathbf{M}_3 = \mathrm{Var}_u[\boldsymbol{\psi}(\widehat{f}, \mathbf{X}; \boldsymbol{\theta}^*)], \mathbf{M}_4 = \mathrm{Cov}_l[\boldsymbol{\psi}(Y, \mathbf{X}; \boldsymbol{\theta}^*), \boldsymbol{\psi}(\widehat{f}, \mathbf{X}; \boldsymbol{\theta}^*)], \mathbf{A}_{\boldsymbol{\theta}^*} = \mathbb{E}[\partial \boldsymbol{\psi}(Y, \mathbf{X}; \boldsymbol{\theta}^*)/\partial \boldsymbol{\theta}], \mathbf{A}_{\boldsymbol{\eta}} = \mathbb{E}[\partial \boldsymbol{\psi}(\widehat{f}, \mathbf{X}; \boldsymbol{\eta})/\partial \boldsymbol{\eta}],$ and $\rho = \dfrac{n}{N}$.

Rewritten $\mathbf{V}(\widehat{\boldsymbol{\theta}}(\boldsymbol{\omega}))$ using the above notation, we have

$$\mathbf{V}(\widehat{\boldsymbol{\theta}}(\boldsymbol{\omega})) = \mathbf{A}_{\boldsymbol{\theta}^*}^{-1}\mathbf{M}_1\mathbf{A}_{\boldsymbol{\theta}^*}^{-1} + \boldsymbol{\omega}^{\mathrm{T}}\mathbf{A}_{\boldsymbol{\eta}}^{-1}\mathbf{M}_2\mathbf{A}_{\boldsymbol{\eta}}^{-1}\boldsymbol{\omega} + \rho\boldsymbol{\omega}^{\mathrm{T}}\mathbf{A}_{\boldsymbol{\eta}}^{-1}\mathbf{M}_3\mathbf{A}_{\boldsymbol{\eta}}^{-1}\boldsymbol{\omega} - 2\boldsymbol{\omega}^{\mathrm{T}}\mathbf{A}_{\boldsymbol{\eta}}^{-1}\mathbf{M}_4\mathbf{A}_{\boldsymbol{\theta}^*}^{-1} \quad (10)$$

$$= \mathbf{A}_{\boldsymbol{\theta}^*}^{-1}\mathbf{M}_1\mathbf{A}_{\boldsymbol{\theta}^*}^{-1} + \boldsymbol{\omega}^{\mathrm{T}}\mathbf{A}_{\boldsymbol{\eta}}^{-1}(\mathbf{M}_2 + \rho\mathbf{M}_3)\mathbf{A}_{\boldsymbol{\eta}}^{-1}\boldsymbol{\omega} - 2\boldsymbol{\omega}^{\mathrm{T}}\mathbf{A}_{\boldsymbol{\eta}}^{-1}\mathbf{M}_4\mathbf{A}_{\boldsymbol{\theta}^*}^{-1} \quad (11)$$

Plug in

$$\boldsymbol{\omega} = \boldsymbol{\omega}_0 = (\mathrm{Var}(\widehat{\boldsymbol{\eta}}_{\mathcal{L}}) + \mathrm{Var}(\widehat{\boldsymbol{\eta}}_{\mathcal{U}}))^{-1} \mathrm{Cov}(\widehat{\boldsymbol{\theta}}_{\mathcal{L}}, \widehat{\boldsymbol{\eta}}_{\mathcal{L}}) \quad (12)$$

$$= (\mathbf{V}(\widehat{\boldsymbol{\eta}}_{\mathcal{L}}) + \rho\mathbf{V}(\widehat{\boldsymbol{\eta}}_{\mathcal{U}}))^{-1}\mathbf{V}(\widehat{\boldsymbol{\theta}}_{\mathcal{L}}, \widehat{\boldsymbol{\eta}}_{\mathcal{L}}) \quad (13)$$

$$= (\mathbf{A}_{\boldsymbol{\eta}}^{-1}\mathbf{M}_2\mathbf{A}_{\boldsymbol{\eta}}^{-1} + \rho\mathbf{A}_{\boldsymbol{\eta}}^{-1}\mathbf{M}_3\mathbf{A}_{\boldsymbol{\eta}}^{-1})^{-1}\mathbf{A}_{\boldsymbol{\eta}}^{-1}\mathbf{M}_4\mathbf{A}_{\boldsymbol{\theta}^*}^{-1} \quad (14)$$

$$= \mathbf{A}_{\boldsymbol{\eta}}(\mathbf{M}_2 + \rho\mathbf{M}_3)^{-1}\mathbf{M}_4\mathbf{A}_{\boldsymbol{\theta}^*}^{-1} \quad (15)$$

into the equation above, we have

$$\mathbf{V}(\widehat{\boldsymbol{\theta}}(\boldsymbol{\omega}_0)) = \mathbf{A}_{\boldsymbol{\theta}^*}^{-1}\mathbf{M}_1\mathbf{A}_{\boldsymbol{\theta}^*}^{-1} - \mathbf{A}_{\boldsymbol{\theta}^*}^{-1}\mathbf{M}_4^{\mathrm{T}}\mathbf{A}_{\boldsymbol{\eta}}^{-1}[\mathbf{A}_{\boldsymbol{\eta}}(\mathbf{M}_2 + \rho\mathbf{M}_3)^{-1}\mathbf{A}_{\boldsymbol{\eta}}]\mathbf{A}_{\boldsymbol{\eta}}^{-1}\mathbf{M}_4\mathbf{A}_{\boldsymbol{\theta}^*}^{-1} \quad (16)$$

$$= \mathbf{A}_{\boldsymbol{\theta}^*}^{-1}\mathbf{M}_1\mathbf{A}_{\boldsymbol{\theta}^*}^{-1} - \mathbf{A}_{\boldsymbol{\theta}^*}^{-1}\mathbf{M}_4^{\mathrm{T}}(\mathbf{M}_2 + \rho\mathbf{M}_3)^{-1}\mathbf{M}_4\mathbf{A}_{\boldsymbol{\theta}^*}^{-1} \quad (17)$$

$$= \mathbf{A}_{\boldsymbol{\theta}^*}^{-1}\left\{\mathbf{M}_1 - \mathbf{M}_4^{\mathrm{T}}(\mathbf{M}_2 + \rho\mathbf{M}_3)^{-1}\mathbf{M}_4\right\}\mathbf{A}_{\boldsymbol{\theta}^*}^{-1}. \quad (18)$$

To connect our protocol with existing methods, we define

$$\boldsymbol{\Sigma}(\boldsymbol{\omega}) = \mathbf{A}_{\boldsymbol{\theta}^*}^{-1}\mathbf{M}_1\mathbf{A}_{\boldsymbol{\theta}^*}^{-1} + \boldsymbol{\omega}^{\mathrm{T}}\mathbf{A}_{\boldsymbol{\theta}^*}^{-1}\mathbf{M}_2\mathbf{A}_{\boldsymbol{\theta}^*}^{-1}\boldsymbol{\omega}^{\mathrm{T}} + \boldsymbol{\omega}^{\mathrm{T}}\rho\mathbf{A}_{\boldsymbol{\theta}^*}^{-1}\mathbf{M}_3\mathbf{A}_{\boldsymbol{\theta}^*}\boldsymbol{\omega} - 2\boldsymbol{\omega}^{\mathrm{T}}\mathbf{A}_{\boldsymbol{\theta}^*}^{-1}\mathbf{M}_4\mathbf{A}_{\boldsymbol{\theta}^*}^{-1} \quad (19)$$

$$\boldsymbol{\omega}_{tr}^* := \arg\min_{\boldsymbol{\omega}_{tr}} \mathrm{Tr}[\boldsymbol{\Sigma}(\boldsymbol{\omega}_{tr})] \text{ where } \boldsymbol{\omega}_{tr} = [\omega_{tr}, \ldots, \omega_{tr}]^{\mathrm{T}} \in \mathbb{R}^K \quad (20)$$

$$\boldsymbol{\omega}_{\mathrm{ele}}^* := [\omega_{\mathrm{ele},1}^*, \ldots, \omega_{\mathrm{ele},K}^*] \in \mathbb{R}^K \text{ where } \omega_{\mathrm{ele,k}}^* = \arg\min_{\omega_{\mathrm{ele,k}}} \boldsymbol{\Sigma}(\boldsymbol{\omega}_{\mathrm{ele}})_{kk} \quad (21)$$

By the theory of PSPA, PPI++, and PPI paper [3, 4, 35], we have

$$n^{1/2}(\widehat{\boldsymbol{\theta}}_{\mathtt{PSPA}} - \boldsymbol{\theta}^*) \xrightarrow{D} \mathcal{N}(\mathbf{0}, \boldsymbol{\Sigma}(\mathrm{diag}(\boldsymbol{\omega}_{\mathrm{ele}}^*))) \quad (22)$$

$$n^{1/2}(\widehat{\boldsymbol{\theta}}_{\mathtt{PPI++}} - \boldsymbol{\theta}^*) \xrightarrow{D} \mathcal{N}(\mathbf{0}, \boldsymbol{\Sigma}(\mathrm{diag}(\boldsymbol{\omega}_{\mathrm{tr}}^*))) \quad (23)$$

$$n^{1/2}(\widehat{\boldsymbol{\theta}}_{\mathtt{PPI}} - \boldsymbol{\theta}^*) \xrightarrow{D} \mathcal{N}(0, \boldsymbol{\Sigma}(\mathrm{diag}(\mathbf{1}))) \quad (24)$$

Based on the proof of Theorem 1, we have the

$$n^{1/2}(\widehat{\boldsymbol{\theta}}(\boldsymbol{\omega}) - \boldsymbol{\theta}^*) \xrightarrow{D} \mathcal{N}\left(0, \mathbf{A}_{\boldsymbol{\theta}^*}^{-1}\mathbf{M}_1\mathbf{A}_{\boldsymbol{\theta}^*}^{-1} + \boldsymbol{\omega}^{\mathrm{T}}\mathbf{A}_{\boldsymbol{\eta}}^{-1}\mathbf{M}_2\mathbf{A}_{\boldsymbol{\eta}}^{-1}\boldsymbol{\omega} + \rho\boldsymbol{\omega}^{\mathrm{T}}\mathbf{A}_{\boldsymbol{\eta}}^{-1}\mathbf{M}_3\mathbf{A}_{\boldsymbol{\eta}}\boldsymbol{\omega} - 2\boldsymbol{\omega}^{\mathrm{T}}\mathbf{A}_{\boldsymbol{\eta}}^{-1}\mathbf{M}_4\mathbf{A}_{\boldsymbol{\theta}^*}^{-1}\right). \quad (25)$$

Plug in $\boldsymbol{\omega}$ with $\mathrm{diag}(\boldsymbol{\omega}_{\mathrm{ele}}^*)A_{\boldsymbol{\theta}^*}^{-1}\mathbf{A}_{\boldsymbol{\eta}}, \mathrm{diag}(\boldsymbol{\omega}_{\mathrm{tr}}^*)A_{\boldsymbol{\theta}^*}^{-1}\mathbf{A}_{\boldsymbol{\eta}},$ and $\mathrm{diag}(\mathbf{1})A_{\boldsymbol{\theta}^*}^{-1}\mathbf{A}_{\boldsymbol{\eta}}$, we have

$$n^{1/2}(\widehat{\boldsymbol{\theta}}(\mathrm{diag}(\boldsymbol{\omega}_{\mathrm{ele}}^*)A_{\boldsymbol{\theta}^*}^{-1}\mathbf{A}_{\boldsymbol{\eta}}) - \boldsymbol{\theta}^*) \xrightarrow{D} \mathcal{N}(\mathbf{0}, \boldsymbol{\Sigma}(\mathrm{diag}(\boldsymbol{\omega}_{\mathrm{ele}}^*))) \quad (26)$$

$$n^{1/2}(\widehat{\boldsymbol{\theta}}(\mathrm{diag}(\boldsymbol{\omega}_{\mathrm{tr}}^*)A_{\boldsymbol{\theta}^*}^{-1}\mathbf{A}_{\boldsymbol{\eta}}) - \boldsymbol{\theta}^*) \xrightarrow{D} \mathcal{N}(\mathbf{0}, \boldsymbol{\Sigma}(\mathrm{diag}(\boldsymbol{\omega}_{\mathrm{tr}}^*))) \quad (27)$$

$$n^{1/2}(\widehat{\boldsymbol{\theta}}(\mathrm{diag}(\mathbf{1})A_{\boldsymbol{\theta}^*}^{-1}\mathbf{A}_{\boldsymbol{\eta}}) - \boldsymbol{\theta}^*) \xrightarrow{D} \mathcal{N}(0, \boldsymbol{\Sigma}(\mathrm{diag}(\mathbf{1}))) \quad (28)$$

Denote $\mathbf{C} = \mathbf{A}_{\boldsymbol{\theta}^*}^{-1}\mathbf{A}_{\boldsymbol{\eta}}$, we have PSPA, PPI++, and PPI is asymptotically equivalent to $\mathrm{diag}(\boldsymbol{\omega}_{\mathrm{ele}}^*)\mathbf{C}, \mathrm{diag}(\boldsymbol{\omega}_{\mathrm{tr}}^*)\mathbf{C},$ and $\mathrm{diag}(\mathbf{1})\mathbf{C}$, respectively. This completes the proof. $\quad\square$

### A.3 Proof of Proposition 2

*Proof.* We apply the first-order Taylor expansion to $g(\widehat{\boldsymbol{\theta}}_{\mathcal{L}}, \widehat{\boldsymbol{\eta}}_{\mathcal{L}}, \widehat{\boldsymbol{\eta}}_{\mathcal{U}})$ around $(\boldsymbol{\theta}^*, \boldsymbol{\eta}, \boldsymbol{\eta})$:

$$g(\widehat{\boldsymbol{\theta}}_{\mathcal{L}}, \widehat{\boldsymbol{\eta}}_{\mathcal{L}}, \widehat{\boldsymbol{\eta}}_{\mathcal{U}}) \approx g(\boldsymbol{\theta}^*, \boldsymbol{\eta}, \boldsymbol{\eta}) + \nabla_{\boldsymbol{\theta}^*}g(\boldsymbol{\theta}^*, \boldsymbol{\eta}, \boldsymbol{\eta})(\widehat{\boldsymbol{\theta}}_{\mathcal{L}} - \boldsymbol{\theta}^*) + \nabla_{\boldsymbol{\eta},1}g(\boldsymbol{\theta}^*, \boldsymbol{\eta}, \boldsymbol{\eta})(\widehat{\boldsymbol{\eta}}_{\mathcal{L}} - \boldsymbol{\eta}) + \nabla_{\boldsymbol{\eta},2}g(\boldsymbol{\theta}^*, \boldsymbol{\eta}, \boldsymbol{\eta})(\widehat{\boldsymbol{\eta}}_{\mathcal{U}} - \boldsymbol{\eta}), \quad (29)$$

where we used $\nabla_{\boldsymbol{\eta},1}$ and $\nabla_{\boldsymbol{\eta},2}$ to denote the gradient of $g(\boldsymbol{\theta}^*, \boldsymbol{\eta}, \boldsymbol{\eta})$ w.r.t the first and second $\boldsymbol{\eta}$, respectively.

This can be written as a linear function of $(\boldsymbol{\theta}^*, \boldsymbol{\eta}, \boldsymbol{\eta})$:

$$g(\widehat{\boldsymbol{\theta}}_{\mathcal{L}}, \widehat{\boldsymbol{\eta}}_{\mathcal{L}}, \widehat{\boldsymbol{\eta}}_{\mathcal{U}}) = \boldsymbol{\mu} + \boldsymbol{\beta}_1^{\mathrm{T}} \widehat{\boldsymbol{\theta}}_{\mathcal{L}} + \boldsymbol{\beta}_2^{\mathrm{T}} \widehat{\boldsymbol{\eta}}_{\mathcal{L}} + \boldsymbol{\beta}_3^{\mathrm{T}} \widehat{\boldsymbol{\eta}}_{\mathcal{U}}, \tag{30}$$

where $\boldsymbol{\mu} = g(\boldsymbol{\theta}^*, \boldsymbol{\eta}, \boldsymbol{\eta}) - \nabla_{\boldsymbol{\theta}^*} g(\boldsymbol{\theta}^*, \boldsymbol{\eta}, \boldsymbol{\eta})\boldsymbol{\theta}^* - 2\nabla_{\boldsymbol{\eta}} g(\boldsymbol{\theta}^*, \boldsymbol{\eta}, \boldsymbol{\eta})\boldsymbol{\eta}$, $\boldsymbol{\beta}_1 = \nabla_{\boldsymbol{\theta}^*} g(\boldsymbol{\theta}^*, \boldsymbol{\eta}, \boldsymbol{\eta})$, $\boldsymbol{\beta}_2 = \nabla_{\boldsymbol{\eta},1} g(\boldsymbol{\theta}^*, \boldsymbol{\eta}, \boldsymbol{\eta})$, $\boldsymbol{\beta}_3 = \nabla_{\boldsymbol{\eta},2} g(\boldsymbol{\theta}^*, \boldsymbol{\eta}, \boldsymbol{\eta})$. Since we require $g(\widehat{\boldsymbol{\theta}}_{\mathcal{L}}, \widehat{\boldsymbol{\eta}}_{\mathcal{L}}, \widehat{\boldsymbol{\eta}}_{\mathcal{U}}) \xrightarrow{P} \boldsymbol{\theta}^*$, we have $\boldsymbol{\mu} = \mathbf{0}$, $\boldsymbol{\beta}_1 = \mathbf{1}$, and $\boldsymbol{\beta}_2 + \boldsymbol{\beta}_3 = \mathbf{0}$. This leads to

$$g(\widehat{\boldsymbol{\theta}}_{\mathcal{L}}, \widehat{\boldsymbol{\eta}}_{\mathcal{L}}, \widehat{\boldsymbol{\eta}}_{\mathcal{U}}) = \widehat{\boldsymbol{\theta}}_{\mathcal{L}} - \boldsymbol{\beta}_3^{\mathrm{T}}(\widehat{\boldsymbol{\eta}}_{\mathcal{U}} - \widehat{\boldsymbol{\eta}}_{\mathcal{L}}), \tag{31}$$

Given

$$n^{1/2} \begin{pmatrix} \widehat{\boldsymbol{\theta}}_{\mathcal{L}} - \boldsymbol{\theta}^* \\ \widehat{\boldsymbol{\eta}}_{\mathcal{L}} - \boldsymbol{\eta} \\ \widehat{\boldsymbol{\eta}}_{\mathcal{U}} - \boldsymbol{\eta} \end{pmatrix} \xrightarrow{D} \mathcal{N} \left\{ \begin{pmatrix} \mathbf{0}_K \\ \mathbf{0}_K \\ \mathbf{0}_K \end{pmatrix}, \begin{pmatrix} \mathbf{V}(\widehat{\boldsymbol{\theta}}_{\mathcal{L}}) & \mathbf{V}(\widehat{\boldsymbol{\theta}}_{\mathcal{L}}, \widehat{\boldsymbol{\eta}}_{\mathcal{L}}) & \mathbf{0} \\ \mathbf{V}(\widehat{\boldsymbol{\theta}}_{\mathcal{L}}, \widehat{\boldsymbol{\eta}}_{\mathcal{L}}) & \mathbf{V}(\widehat{\boldsymbol{\eta}}_{\mathcal{L}}) & \mathbf{0} \\ \mathbf{0} & \mathbf{0} & \rho\mathbf{V}(\widehat{\boldsymbol{\eta}}_{\mathcal{U}}) \end{pmatrix} \right\}, \tag{32}$$

we have

$$\mathbf{V}(g(\widehat{\boldsymbol{\theta}}_{\mathcal{L}}, \widehat{\boldsymbol{\eta}}_{\mathcal{L}}, \widehat{\boldsymbol{\eta}}_{\mathcal{U}})) = \mathbf{V}(\widehat{\boldsymbol{\theta}}_{\mathcal{L}}) + \boldsymbol{\beta}_3^{\mathrm{T}} \mathbf{V}(\widehat{\boldsymbol{\eta}}_{\mathcal{L}})\boldsymbol{\beta}_3 + \boldsymbol{\beta}_3^{\mathrm{T}} \rho \mathbf{V}(\widehat{\boldsymbol{\eta}}_{\mathcal{U}})\boldsymbol{\beta}_3 - 2\boldsymbol{\beta}_3^{\mathrm{T}} \mathbf{V}(\widehat{\boldsymbol{\theta}}_{\mathcal{L}}, \widehat{\boldsymbol{\eta}}_{\mathcal{L}}), \tag{33}$$

which is a quadratic function of $\boldsymbol{\beta}_3$ and achieves it minimum when $\boldsymbol{\beta}_3 = (\mathbf{V}(\widehat{\boldsymbol{\eta}}_{\mathcal{L}}) + \rho\mathbf{V}(\widehat{\boldsymbol{\eta}}_{\mathcal{U}}))^{-1}\mathbf{V}(\widehat{\boldsymbol{\theta}}_{\mathcal{L}}, \widehat{\boldsymbol{\eta}}_{\mathcal{L}}) = \boldsymbol{\omega}_0$. This completes the proof. $\square$

## B An example for understanding the difference between PSPS and PPI++

Consider a linear regression with two predictors: $Y \sim \theta_1 X_1 + \theta_2 X_2$. The summary statistics for PSPS can be expressed as: $\left[\hat{\theta}_{1\mathcal{L}}, \hat{\theta}_{2\mathcal{L}}, \hat{\eta}_{1\mathcal{L}}, \hat{\eta}_{2\mathcal{L}}, \hat{\eta}_{1\mathcal{U}}, \hat{\eta}_{2\mathcal{U}}\right]^T$ from linear regression analysis in labeled and unlabeled data.

For PSPS, since $\hat{\theta}_{\text{PSPS}} = \begin{bmatrix} \hat{\theta}_{1\mathcal{L}} \\ \hat{\theta}_{2\mathcal{L}} \end{bmatrix} - \begin{bmatrix} w_1 & w_{12} \\ w_{12} & w_2 \end{bmatrix} \begin{bmatrix} \hat{\eta}_{1\mathcal{L}} \\ \hat{\eta}_{2\mathcal{L}} \end{bmatrix} + \begin{bmatrix} w_1 & w_{12} \\ w_{12} & w_2 \end{bmatrix} \begin{bmatrix} \hat{\eta}_{1\mathcal{U}} \\ \hat{\eta}_{2\mathcal{U}} \end{bmatrix}$, the final estimator for $\theta_1$ is $\hat{\theta}_{\text{PSPS},1} = \hat{\theta}_{1\mathcal{L}} - w_1\hat{\eta}_{1\mathcal{L}} + w_1\hat{\eta}_{1\mathcal{U}} - w_{12}\hat{\eta}_{2\mathcal{L}} + \omega_{12}\hat{\eta}_{2\mathcal{U}}$.

In comparison, since $\hat{\theta}_{\text{PPI++}} = \begin{bmatrix} \hat{\theta}_{1\mathcal{L}} \\ \hat{\theta}_{2\mathcal{L}} \end{bmatrix} - \begin{bmatrix} w & 0 \\ 0 & w \end{bmatrix} \begin{bmatrix} \hat{\eta}_{1\mathcal{L}} \\ \hat{\eta}_{2L} \end{bmatrix} + \begin{bmatrix} w & 0 \\ 0 & w \end{bmatrix} \begin{bmatrix} \hat{\eta}_{1\mathcal{U}} \\ \hat{\eta}_{2U} \end{bmatrix}$, its estimator for $\theta_1$ is $\hat{\theta}_{\text{PPI++},1} = \hat{\theta}_{1\mathcal{L}} - w\hat{\eta}_{1\mathcal{L}} + w\hat{\eta}_{1\mathcal{U}}$.

Since $\hat{\theta}_{\text{PSPS},1}$ involves two zero-augmentation terms (i.e., $-w_1\hat{\eta}_{1\mathcal{L}} + w_1\hat{\eta}_{1U}$ and $-\omega_{12}\hat{\eta}_{2\mathcal{L}} + \omega_{12}\hat{\eta}_{2\mathcal{U}}$), its asymptotic variance should be less than or equal to that of PPI++ with one augmentation term. Therefore, PSPS borrows information from both coordinates, but PPI++ is restricted to information from only the first coordinate. Although the PPI++ can be used under a different scalarization, it still contains one augmentation term.

## C Algorithms for ML-assisted FDR control

---

**Algorithm 2** PSPS-BH for linear regression

---

**Input:** Labeled data $\mathcal{L} = (\mathbf{X}_{\mathcal{L}}, Y_{\mathcal{L}}, \widehat{f}_{\mathcal{L}})$, unlabeled data $\mathcal{U} = (\mathbf{X}_{\mathcal{U}}, \widehat{f}_{\mathcal{U}})$, FDR level $q \in (0, 1)$.
1: Obtain the p-value $p_k$ for features $k = 1, \cdots, K$ by ML-assisted linear regression PSPS-LR($\mathcal{L}, \mathcal{U}$)
2: Sort the p-values in ascending order $p_{(1)} \leq p_{(2)} \leq \ldots \leq p_{(K)}$
3: Finds the p-value cutoff $\tau_q^{\text{BH}} := p_{(k)}$, where $k = \max\left\{k = 1, \ldots, K : p_{(k)} \leq \frac{k}{K}q\right\}$
**Output:** Discoveries $\widehat{S} = \left\{k : p_k \leq \tau_q^{\text{BH}}\right\}$

---

---

**Algorithm 3** PSPS-knockoff with debiased Lasso

---

**Input:** Labeled data $\mathcal{L} = (\mathbf{X}_{\mathcal{L}}, Y_{\mathcal{L}}, \widehat{f}_{\mathcal{L}})$, unlabeled data $\mathcal{U} = (\mathbf{X}_{\mathcal{U}}, \widehat{f}_{\mathcal{U}})$, FDR level $q \in (0, 1)$.

1: Obtain the augmented labeled and unlabeled data as $\tilde{\mathcal{L}} = (\mathbf{X}_{\mathcal{L}}, \tilde{\mathbf{X}}_{\mathcal{L}}, Y_{\mathcal{L}}, \widehat{f}_{\mathcal{L}})$ and $\tilde{\mathcal{U}} = (\mathbf{X}_{\mathcal{U}}, \tilde{\mathbf{X}}_{\mathcal{U}}, \widehat{f}_{\mathcal{U}})$ where $\tilde{\mathbf{X}}_{\mathcal{L}} \leftarrow \texttt{knockoff-sample}(\boldsymbol{X}_{\mathcal{L}})$ and $\tilde{\mathbf{X}}_{\mathcal{U}} \leftarrow \texttt{knockoff-sample}(\boldsymbol{X}_{\mathcal{U}})$.

2: Calculate the PSPS debiased Lasso coefficient $\widehat{\boldsymbol{\beta}}^{\texttt{PSPS-DLasso}} \leftarrow \texttt{PSPS-DLasso}(\tilde{\mathcal{L}}, \tilde{\mathcal{U}})$

3: $W_k$ for $k = 1, \cdots, K \leftarrow \texttt{knockoff-statistic}(\widehat{\boldsymbol{\beta}}^{\texttt{PSPS-DLasso}})$

4: Set the cutoff $\tau_q^{\text{knockoff}} = \left\{ t > 0 : \frac{\#\{k : W_k \leq -t\}}{\#\{k : W_k \geq t\} \vee 1} \leq q \right\}$

**Output:** Discoveries $\widehat{S} = \left\{ k : M_k > \tau_q^{\text{knockoff}} \right\}$

---

Here, we employ second-order multivariate Gaussian knockoff variables for `knockoff-sample` and use the difference between the absolute values of the $k$-th feature and its knockoff coefficient as the `knockoff-statistic`. Alternative choices for these two steps can also be readily integrated into our algorithm [26].

## D  Implementation details

### D.1  Simulation

Here, we provide the details for our simulation. All our simulation is run in R with version 4.2.1 (2022-06-23) in a MacBook Air with an M1 chip. For all the simulations, the ground truth coefficients are obtained using $5 \times 10^4$ samples; A pre-trained random forest with 500 trees to grow is obtained from hold-out data. We bootstrap the labeled data for 200 times for covariance estimation. All simulations are repeated 1000 times.

- **Mean estimation**, **Linear regression**, and **Quantile regression**: We simulate the data from $Y_i = X_{1i}\beta_1 + X_{2i}\beta_2 + X_{1i}^2\beta_2 + X_{2i}^2\beta_2 + \epsilon_i$, where $X_{i1}$ and $X_{2i}$ are independent simulated from $\mathcal{N}(0, 1)$, $\beta_1 = \sqrt{0.08}$, $\beta_2$ is set to be the value such that $X_{2i}\beta_2 + X_{1i}^2\beta_2 + X_{2i}^2\beta_2$ explains 81% of the outcome variance, and $\epsilon_i$ is simulated from a mean zero normal distribution with variance such that $\text{Var}(Y_i) = 1$. We use $X_{1i}$ and $X_{2i}$ as features to predict the $Y_i$ in the random forest. We consider labeled data with 500 individuals, and unlabeled data with sample size $1000, 2500, 5000$, or $10000$.

- **Logistic regression**: We simulate the data from $Y_i = \mathbb{1}(\tilde{Y}_i > \text{median}\,(\tilde{Y}_i))$, where $\tilde{Y}_i = X_{1i}\beta_1 + X_{2i}\beta_2 + X_{1i}^2\beta_2 + X_{2i}^2\beta_2 + \epsilon_i$, where $X_{i1}$ and $X_{2i}$ are independent simulated from $\mathcal{N}(0, 1)$, $\beta_1 = \sqrt{0.08}$, $\beta_2$ is set to be the value such that $X_{2i}\beta_2 + X_{1i}^2\beta_2 + X_{2i}^2\beta_2$ explains 81% of the outcome variance, and $\epsilon_i$ is simulated from a mean zero normal distribution with variance such that $\text{Var}(\tilde{Y}_i) = 1$. We use $X_{1i}$ and $X_{2i}$ as features to predict the $Y_i$ in the random forest. We consider labeled data with 500 individuals, and unlabeled data with sample size $1000, 2500, 5000$, or $10000$.

- **Instrumental variable (IV) regression**: We simulate the data by

$$Z_i \sim \mathcal{N}(0, 1), \tag{34}$$
$$X_{1i} = 0.4Z_i + \delta_i, \delta_i \sim \mathcal{N}(0, 0.84), \tag{35}$$
$$X_{2i} = 0.3Z_i + 0.8Y_i + \gamma_i, \text{ where } \gamma_i \sim \mathcal{N}(0, \tau_\gamma), \text{ such that } \text{Var}(X_{2i}) = 1, \tag{36}$$
$$Y_i = \sqrt{0.08}X_{1i} + \epsilon_i, \text{ where } \epsilon_i \sim \mathcal{N}(0, \tau_\epsilon), \text{ such that } \text{Var}(Y_i) = 1 \tag{37}$$

  We use $X_{1i}$ and $X_{2i}$ as features to predict the $Y_i$ in the random forest. We consider labeled data with 500 individuals, and unlabeled data with sample size $1000, 2500, 5000$, or $10000$. The $Z_i$ is used as a instrument for $X_{1i}$.

- **Negative binomial (NB) regression**: We simulate the data by

$$X_{1i} \sim \mathcal{N}(0, 1), X_{2i} \sim \mathcal{N}(0, 1), \tag{38}$$
$$\mu_i = \exp(\sqrt{0.3}X_{1i} + 0.8X_{2i}) \tag{39}$$
$$Y_i = \text{NegativeBinomial}\,(s = k, \mu = \mu_i)\,, \text{ where } s \text{ is the dispersion parameter and } \mu \text{ is the rate.} \tag{40}$$

We use $X_{1i}$ and $X_{2i}$ as features to predict the $Y_i$ in the random forest. We consider labeled data with 500 individuals, and unlabeled data with sample size $1000, 2500, 5000$, or $10000$.

- **Debiased Lasso**: We simulate the data by

$$X_{1i}, \ldots, X_{200i} \sim \mathcal{N}(0, 1) \tag{41}$$

$$\theta_1, \ldots, \theta_{15} = \frac{0.9}{\sqrt{15}}; \theta_{16}, \ldots, \theta_{200} = 0 \tag{42}$$

$$Y_i = \sum_{k=1}^{150} X_{ki}\theta_k + \epsilon_i, \epsilon_i \sim \mathcal{N}(0, \tau_\epsilon) \text{ such that } \mathrm{Var}(Y_i) = 1. \tag{43}$$

We use $X_{1i}, \ldots, X_{200i}$ as features to predict the $Y_i$ in the random forest. We consider labeled data with 100 individuals, and unlabeled data with sample size $1500, 2000, 2500$, or $3000$.

- **Wilcoxon rank-sum test**: We simulate the data by

$$X_{1i} \sim \mathrm{Bernoulli}(0.5), X_{2i} \sim \mathcal{N}(0, 1) \tag{44}$$

$$Y_i = \beta_1 X_{1i} + \beta_2 X_{2i} + \beta_2 X_{2i}^2 + \epsilon_i, \epsilon_i \sim \mathcal{N}(0, \tau_\epsilon), \tag{45}$$

where $\beta_1 = \sqrt{0.01}$ to power simulation and $\beta_1 = 0$ for type-I error simulation, $\beta_2$ is set to be the value such that $\beta_2 X_{2i} + \beta_2 X_{2i}^2$ explains $81\%$ of the outcome variance, and $\tau_\epsilon$ is set to be the value such that $\mathrm{Var}(Y_i) = 0$. We use $X_{1i}$ and $X_{2i}$ as features to predict the $Y_i$ in the random forest. We consider labeled data with 500 individuals, and unlabeled data with sample size $1000, 2500, 5000$, or $10000$.

- **Benjamini-Hochberg (BH) procedure**: We set $K = 150$ generate the features independently from $\mathcal{N}(0, \Sigma)$, where $\Sigma$ is a symmetric Toeplitz matrix that has the structure:

$$\Sigma = \begin{bmatrix} r^0 & r^1 & \ldots & r^{p-1} \\ r^1 & \ddots & \ldots & r^{p-2} \\ \vdots & \ldots & \ddots & \vdots \\ r^{p-1} & r^{p-2} & \ldots & r^0 \end{bmatrix} \tag{46}$$

The correlation $r$ is set to be $0.25$. We then simulate the outcome $Y_i = \sum_{k=1}^{150} X_{ki}\beta_k + \epsilon_i$, where we randomly sample 15 $\beta_k$ to be $0.15$ and let all other remaining $\beta_k$ to be 0. $\epsilon_i$ is simulated from a mean-zero normal distribution with variance set to the value such that $\mathrm{Var}(Y_i) = 1$. We further generate $Z_i = 0.9 Y_i + \sum_{k=1}^{150} X_{ki}\theta_k + \gamma_i$, where $\theta_k = 0.15$ for all $k = 1, \ldots, 150$. We use $Z_i$ as features to predict the $Y_i$ in the random forest. We consider labeled data with 500 individuals, and unlabeled data with sample size 5000.

- **knockoff**: We used the same setting as described in the BH procedure above to generate the data. The only difference is that we set $\beta_k = 0.5$ for features associated with the outcome and considered labeled data consisting of 100 individuals, along with unlabeled data comprising a sample size of 1000.

### D.2 Identify vQTLs for bone mineral density

Our prediction pipeline comprises two components: prediction for unlabeled data and prediction for labeled data. To predict bone mineral density in unlabeled data, we first selected predictive features by 1) calculating the correlation of bone mineral density with 466 other variables (sample size > 200,000 from UK Biobank) using labeled data and 2) selecting the top 50 variables with the highest correlations as inputs for the SoftImpute algorithm [30] to predict bone mineral density in the unlabeled data. For the labeled data, we employ a similar approach but incorporate 10-fold cross-validation to prevent overfitting. We select the predictive variables and train the SoftImpute model using 90% of the labeled data. We then perform predictions on the remaining 10% in each fold and repeat this process 10 times across all folds.

# E   Supplementary figures and tables

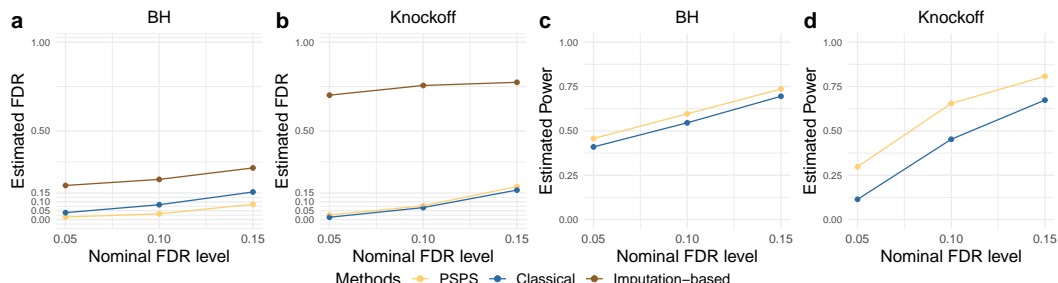

Figure E.1: Simulation for FDR control. Panel a-b shows the estimated FDR level given the expected FDR. Panel c-d shows the power.

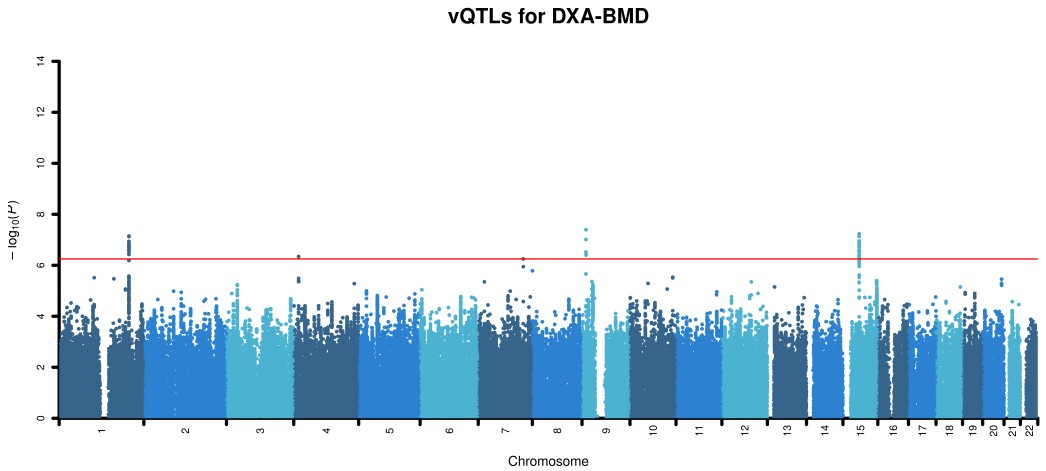

Figure E.2: Manhattan plot of vQTLs for bone mineral density. The X-axis represents chromosomes (CHR), plotted by base pair positions (BP). Each point on the plot indicates a single nucleotide polymorphism (SNP). The Y-axis depicts -log10(p-values).

| CHR | BP | SNP | A1 | A2 | EAF | BETA | SE | p-value | FDR |
|-----|----|-----|----|----|----|------|----|---------|-----|
| 1 | 205359339 | rs12139623 | T | C | 0.105 | 0.063 | 0.012 | 7.2e-08 | 0.033 |
| 4 | 14359045 | rs552582509 | A | G | 0.213 | 0.046 | 0.009 | 4.5e-07 | 0.045 |
| 7 | 132568586 | rs79089873 | A | G | 0.073 | -0.068 | 0.014 | 5.6e-07 | 0.049 |
| 9 | 11587905 | rs146938822 | A | G | 0.022 | -0.134 | 0.024 | 4.0e-08 | 0.033 |
| 15 | 47672201 | rs281258 | C | T | 0.393 | -0.04 | 0.007 | 5.8e-08 | 0.033 |

Table E.1: Significant vQTLs for bone mineral density. Abbreviations: CHR, Chromosome; BP, Base Pair; SNP, Single Nucleotide Polymorphism; A1, Allele 1 (Effect Allele); A2, Allele 2 (Non-effect Allele); EAF, Effect Allele Frequency; BETA, Effect Size (Beta Coefficient); SE, Standard Error; FDR, False Discovery Rate.

| Method | Linear regression | Logistic regression |
|--------|-------------------|---------------------|
| PSPS   | 1.62s             | 8.27s               |
| PPI    | 0.024s            | 0.032s              |
| PPI++  | 0.031s            | 0.077s              |
| PSPA   | 0.049s            | 0.034s              |

Table E.2: Runtime experiments. Utilizing a dataset with 500 labeled and 10,000 unlabeled data points, PSPS required 1.62 seconds for linear regression and 8.27 seconds for logistic regression using 200 bootstrap resampling. The computation of one-step debiasing using summary statistics alone took 0.032 seconds for linear regression and 0.033 seconds for logistic regression. Current methods, which estimate asymptotic variance via the closed form derived by the Central Limit Theorem instead of resampling, ranged from 0.024 to 0.049 seconds for linear regression and 0.032 to 0.077 seconds for logistic regression.

