# OpenReview forum: "Task-Agnostic Machine-Learning-Assisted Inference"
_NeurIPS.cc/2024/Conference — NeurIPS 2024 poster_

### Official Review · Reviewer_5S3y · 2024-07-10

**Soundness:** 3
**Presentation:** 2
**Contribution:** 2
**Rating:** 6
**Confidence:** 2

**Summary:**

This paper proposes a post-prediction inference solution (PSPS) that can be adapted into various established data analysis routine and delivers valid and efficient inference for most ML models. In particular, the paper uses both labelled and unlabelled data to derive estimators that are consistent and efficient and only utilizes 1st and 2nd order summary statistics.

**Strengths:**

The paper provides a well-rounded analysis on the proposed method with sensitivity analysis on distributions shifts, to violations of independence assumption between labelled and unlabelled data and to better statistical power in false discovery control.

**Weaknesses:**

- The paper claims the proposed method offers better statistical power in various statistical tasks from mean estimation to quantile regression through various numerical experiments in Section 5.1. However, it is unclear how much of the statistical power is offered by the nature of transductive inference and how much is offered by the proposed protocol. For example, [1] shows that incorporating unlabelled data achieves lower test error than purely labelled data already. Could the author demonstrate the unique advantages offers by the proposed method?
- The protocol proposed relies on assumptions on 1) i.i.d. distributions between labelled and unlabelled data and 2) finding an algorithm applied to labeled data returns a consistent and asymptotic normally distributed estimators. I am wondering how easy is it to find an algorithm that can produce such estimators?
- The authors list one of the key features of the method is privacy-preserving, but did not discuss this point in the remaining paper.

[1] Chapelle, O., Vapnik, V., & Weston, J. (1999). Transductive inference for estimating values of functions. Advances in Neural Information Processing Systems, 12.

**Questions:**

Please see above.

**Limitations:**

Yes.

---

> ### Author Rebuttal · Authors · 2024-08-06
>
> **W1:**
> Reference [1] focuses on using unlabeled data to improve the estimation of regression function (i.e. E[Y|X]). This is a classic machine learning prediction problem which leverages semi-supervised learning to enhance prediction accuracy. In contrast, our study focuses on improving estimation and statistical inference using unlabeled data, emphasizing the need for estimator consistency and the validity of confidence intervals. This is a very new topic (`Angelopoulos, Anastasios N., et al. "Prediction-powered inference." Science (2023)`) in the machine learning field. Because of this, we believe our paper and [1] address fundamentally different problems. In real-world scientific applications, many issues pertain to statistical inference (e.g., estimating the effects of genetic variants on height). The methods in [1] are not suitable for such applications, whereas our methods are specifically designed to tackle these issues.
>
> In our initial submission, we demonstrated the validity and superior statistical efficiency of our method over existing approaches (Section 3.2), substantiated by theoretical guarantees:
> 1. Theorem 1: PSPS achieves element-wise asymptotic variance reduction compared to classical statistical inference (based solely on labeled data) while ensuring estimator consistency.
> 2. Theorem 1: PSPS guarantees valid inference (accurate confidence interval coverage) compared to imputation-based inference (which relies solely on machine learning predictions using unlabeled data).
> 3. Propositions 1 and 2: PSPS has no larger asymptotic variance than existing ML-assisted inference methods that provide consistent estimators.
>
> Furthermore, a key feature of our method is its independence from task-specific derivations and implementations, allowing it to address a broader range of statistical problems compared to existing methods.
> * For M-estimation tasks, currently, only mean and quantile estimation, as well as linear, logistic, and Poisson regression, have been implemented in software and are ready for immediate application. For other M-estimation tasks, task-specific derivation of the ML-assisted loss functions and asymptotic variance are necessary. Next, researchers still need to develop software packages to carry out real applications. In contrast, PSPS only requires already implemented software designed for classical inference using labeled data.
> * For problems that are not considered M-estimation but have asymptotically normally distributed estimators, only PSPS can be applied and all current methods would fail. The principles that facilitate ML-assisted M-estimation are not applicable for these non-M-estimation tasks.
>
> To summarize, our method, designed for estimation and statistical inference, addresses a different problem than [1], which focuses on prediction. We have also demonstrated the substantial advantages of our approach compared to existing methods for ML-assisted statistical inference through both theoretical and experimental analyses. We have also added the semi-supervised learning literature in the related work to avoid confusion.
>
> **W2:**
> Consistent and asymptotically normally distributed estimators are the fundamental properties of an estimator in statistical sciences, and significant research efforts have been devoted to developing such estimators for a wide range of statistical tasks. Generally, if a statistical task allows the application of the Central Limit Theorem. the resulting estimator is likely to be asymptotically normal. Specifically, M-estimator that calculates an estimator through empirical risk minimization, and U-statistics that can be expressed as averages of a symmetric function applied to subsets of a sample are both consistent and asymptotically normally distributed estimators. Common examples of M-estimators include but are not restricted to maximum likelihood estimation, mean and median estimations, linear regression models, generalized linear models, and quantile regression. U-statistics include but is not restricted to variance estimation and Wilcoxon-Rank Sum statistics. In summary, it is fairly straightforward to identify an algorithm capable of producing such estimators for a wide range of statistical tasks.
>
> **W3:**
> The “privacy-preserving" feature of PSPS refers to the fact that we only require summary statistics as input for inference, rather than individual-level raw data (features X and label Y). This terminology is commonly used in human genetics, healthcare, and multi-center electronic health record analysis to describe methods that enable statistical inference without directly accessing personal data. This approach is analogous to federated learning (`Kairouz, Peter, et al. 2021`) in the machine learning literature. For example, consider a scenario where labeled data is in one center and unlabeled data in another, yet researchers cannot access individual-level data from both centers simultaneously. Under such conditions, current ML-assisted inference, which relies on accessing both labeled and unlabeled data to minimize a joint loss function, is not feasible. However, PSPS circumvents this issue by aggregating summary statistics from multiple centers, thus performing statistical inference while upholding the privacy of individual-level data.
>
> We also acknowledge that in the machine learning literature, "privacy-preserving" often specifically refers to techniques like differential privacy. To avoid confusion, we have revised the terminology from “privacy-preserving” to "federated inference" and provided an example to illustrate this use case more clearly.
>
> **Final note**: We are excited that you find our methods achieve better statistical power compared with current methods and our strategy to deal with distributional shifts. If you have any further questions, please do not hesitate to let us know. If our responses have resolved your concerns, we kindly request you to consider increasing your score and championing our paper.

---

> > ### Comment · Reviewer_5S3y · 2024-08-11
> >
> > W1: Thank you for the clarification. I agree that the current work addresses different problems with [1].
> >
> > W2: I agree consistent and asymptotically normal estimators are common in statistical sciences but not convinced it is common for current ML algorithms, e.g., LLM, given the paper focuses on ML-assisted inference.
> >
> > W3: Thank you for the clarification.
> >
> > In general, I agree that this paper proposes a new solution that improves statistical consistency and correct confidence coverage using unlabeled data, however I am not certain the impact of contribution given I am not an expert in this area. Therefore I will raise my score but lower my confidence to reflect this.

---

> ### Author Response · Authors · 2024-08-11
>
> Thanks for your reply!
>
> W2: We would like to clarify that our framework does not require the consistent and asymptotically normal estimators for ML algorithms that are commonly expected in statistics. In fact, our approach can accommodate any "black box" ML algorithm. In our setting, the ML algorithm is used to impute (predict) labels in unlabeled data. These predictions are then used as input to a statistical method (algorithm), such as linear regression, to solve a statistical problem, such as estimating the effect of DNA on height. The requirement for consistent and asymptotically normal estimators applies only to the statistical algorithm, not to the ML algorithm used for label prediction.
>
> We have updated our paper to explicitly state that we did not impose any constraints on the ML algorithms used for label prediction.

---

> > ### Comment · Reviewer_5S3y · 2024-08-11
> >
> > Thank you for addressing the concern on the proposed method's applicability - I am raising the score accordingly while maintaining low confidence.

---

### Official Review · Reviewer_DsrD · 2024-07-11

**Soundness:** 1
**Presentation:** 3
**Contribution:** 3
**Rating:** 7
**Confidence:** 3

**Summary:**

This paper proposes a new unified framework for ML-assisted inference that reduces the general problem to essentially one of estimating normal means, and then applies simple operations (and a bootstrap step) to solve the normal means problem. In addition to the unifying framework’s simplicity, a key result is that the asymptotic efficiency of the proposed estimator dominates that of existing works under mild conditions.

**Strengths:**

1. The problem is an important one
2. The main idea presented in section 2.2 is quite compelling, reducing the ML-assisted inference problem into basically a means estimation problem, and also finding the optimal weighting to combine the components
3. The results about dominating existing approaches in terms of statistical efficiency are exciting

**Weaknesses:**

1. Theorem 1 doesn’t give asymptotically valid inference, since it doesn’t address variance estimation. If the asymptotic variance were known, this result would be sufficient, but it is not, and what is used instead is an estimate of the variance, which must be proved to be consistent in order to conclude asymptotic validity of Algorithm 1’s confidence interval and p-value. This oversight is relevant in other areas, such as eq (35) in the appendix is unjustified and does not follow from eq (34).
2. Prop 1 as stated does not say that these methods all have the same asymptotic variance, which seems to be the implication of the rest of section 3.2 after Prop 1. It seems the proof does prove the right thing, it’s just the statement is wrong (it is missing \sqrt{n} multiplying the LHS of each of the three limits).
3. Proof of prop 4 seems to just be rehashing the proof of knockoffs, except for a single sentence in lines 531-532 which simply states, without proof, the most important part, and the only part that involves the proposed method.

In my opinion, the paper cannot be published without these issues addressed, hence my current score. Item 2 seems to just be a typo, but item 1 is absolutely critical and central, and item 3 is just not rigorous, so it should either be fixed or deleted (personally, I don’t think section 4 strengthens the paper much, since it’s just applying known FDR control techniques to the output of the proposed method, so the validity should just follow from the validity of the proposed method and that of the known FDR control techniques). If they are addressed (and no other reviewers raise other issues that cause me concern), my score would go up quite a bit, as I think this paper has significant strengths.

**Questions:**

In what sense is your method more task-agnostic than others? It seems to still require \mathcal{A}, which is task-specific. The authors mention in lines 129-131 that works have been developed that are general to M-estimators, but many (all?) the simulation tasks in Figure 3 which claim to be ones which “have not been implemented for ML-assisted inference” are M-estimation tasks (or reduce to them), so why couldn’t other methods have been applied to them? Explaining this more clearly would raise my score.

**Limitations:**

The authors have not addressed computational efficiency, which is a critical property—how fast is it relative to the other methods under consideration? Addressing this would raise my score.

---

> ### Author Rebuttal · Authors · 2024-08-06
>
> **W1:**
> Thank you for the comment. We have addressed the variance estimation in Theorem 1 by adding
>
> "
> With $\hat{\mathbf{V}}(\hat{\theta} _{\mathrm{PSPS}}) \xrightarrow{P} \mathbf{V}(\hat{\theta} _{\mathrm{PSPS}}), \lim _n \mathbb{P}(\theta _k^* \in \mathcal{C} _{\alpha, k}^{\mathrm{PSPS}})=1-\alpha$ where $\mathcal{C} _{\alpha, k}^{\mathrm{PSPS}}=(\hat{\boldsymbol{\theta}} _{\mathrm{PSPS} _k} \pm z _{1-\alpha / 2} \sqrt{\hat{\mathrm{V}}(\hat{\theta} _{\mathrm{PSPS}, \mathrm{k}}) / n})$.
> "
>
> Here $\hat{\mathbf{V}}(\hat{\boldsymbol{\theta}} _{\text {PSPS}})=\hat{\mathbf{V}}(\hat{\boldsymbol{\theta}} _{\mathcal{L}})-\hat{\mathbf{V}}(\hat{\boldsymbol{\theta}}
>  _{\mathcal{L}}, \hat{\boldsymbol{\eta}} _\mathcal{L})^{\mathrm{T}}(\hat{\mathbf{V}}(\hat{\boldsymbol{\eta}}  _\mathcal{L})+\rho \hat{\mathbf{V}}(\hat{\boldsymbol{\eta}} _\mathcal{U}))^{-1} \hat{\mathbf{V}}(\hat{\boldsymbol{\theta}} _\mathcal{L}, \hat{\boldsymbol{\eta}}  _\mathcal{L})$ can be obtained by applying the algebraic form of  $\mathbf{V}(\hat{\boldsymbol{\theta}} _{\text {PSPS}})$ using the bootstrap estimators of $\mathbf{V}\(\widehat{\boldsymbol{\theta}} _{\mathcal{L}}\), \mathbf{V}(\hat{\eta} _{\mathcal{L}}), \mathbf{V}(\widehat{\boldsymbol{\theta}} _{\mathcal{L}}, \hat{\eta} _{\mathcal{L}}) \text {, and } \mathbf{V}(\hat{\eta} _{\mathcal{U}})$. Since bootstrap estimators of variance is consistent under certain regularity conditions, by Slutsky’s theorem, $\hat{\mathbf{V}}(\hat{\boldsymbol{\theta}} _{\text {PSPS}})$ is consistent for $\mathbf{V}(\hat{\boldsymbol{\theta}} _{\text {PSPS}})$.
>
> **W2:**
> We have added $\sqrt{n}$ to Prop1. The new Prop 1 is
> $$n^{\frac{1}{2}}\left(\widehat{\boldsymbol{\theta}}\left(\operatorname{diag}\left(\boldsymbol{\omega} _{\text {ele }}\right) \mathbf{C}\right)-\widehat{\boldsymbol{\theta}} _{\text{POP-Inf}}\right) \xrightarrow{D} \mathbf{0},
> n^{\frac{1}{2}}\left(\widehat{\boldsymbol{\theta}}\left(\operatorname{diag}\left(\boldsymbol{\omega} _{\text {tr }}\right) \mathbf{C}\right)-\widehat{\boldsymbol{\theta}} _{\text{PPI}++}\right) \xrightarrow{D} \mathbf{0},
> n^{\frac{1}{2}}\left(\widehat{\boldsymbol{\theta}}\left(\operatorname{diag}\left(\bf{1}\right) \mathbf{C}\right)-\widehat{\boldsymbol{\theta}} _{\text{PPI}}\right) \xrightarrow{D} \mathbf{0},
> $$
> which says these methods all have the same asymptotic variance.
>
> **W3:**
> The statement in lines 531-532 can be proved below: For null $k$, since debiased lasso is consistent, the $k$-th debiased Lasso coefficient converges to 0. Therefore, $W_k$ is (asymptotically) symmetric around 0. Therefore, $Z_k=1(W_k)$ (asymptotically) follows the Bernoulli distribution with a probability of 0.5. There is a typo in the original proof though, that only requires the $Z_k$ (asymptotically) follows Bernoulli(0.5) for null feature $k (\beta_k=0)$.
>
> **Overall:**
> We really appreciate the reviewer for spotting these issues in our original submission. We have addressed these issues in details above. We have also changed Section 4 to a remark in Section 3 to indicate that the results for PSPS can be directly combined with existing FDR control techniques to achieve ML-assisted FDR control.
>
> **Q:**
> Among the tasks illustrated in Figure 3, quantile regression and negative binomial regression are M-estimation problems, and the principles for applying ML-assisted inference to these tasks are available although no specific derivations and software implementations have been made available for broader use. Instrumental variable (IV) regression, while technically an M-estimation task, is typically solved using two-stage least squares, which are non-trivial to implement under a ML-assisted inference framework. debiased Lasso and Wilcoxon Rank-Sum test do not conform to minimizing a loss function. Hence, mathematical principles for the ML-assisted inference of these tasks are still underdeveloped, with no existing software implementations.
>
> PSPS is more task-agnostic than other methods in three aspects:
> * For M-estimation tasks
>   * Current methods: Currently, only mean and quantile estimation, as well as linear, logistic, and Poisson regression, have been implemented in software tools and are ready for immediate application. For other M-estimation tasks, task-specific derivation of the ML-assisted loss functions and asymptotic variance via the central limit theorem are necessary. After that, researchers still need to develop software packages and optimization algorithms to carry out real applications.
>   * In contrast, PSPS only requires already implemented algorithms and software designed for classical inference using labeled data. For example, implementing negative binomial regression with PSPS is straightforward using existing functions:
> ```
> from statsmodels.formula.api import glm
> model = glm('count ~ x1 + x2', data=df, family=NegativeBinomial()).fit()
> ```
> in python
>
> or
> ```
> library(MASS)
> glm.nb(count ~ x1 + x2, data = data)
> ```
> in R
>
> * For problems that are not considered M-estimation but have asymptotically normally distributed estimators, only PSPS can be applied and all current methods would fail. The principles that facilitate ML-assisted M-estimation are not applicable for these non-M-estimation tasks.
>
> **Due to the character limit, we have placed our response to the remaining concerns in the comment section of the reviews.**

---

> > ### Comment · Reviewer_DsrD · 2024-08-10
> >
> > W1: I agree that IF the variance estimator is consistent, then everything works out easily via Slutsky's--this is not the issue. The main issue is that consistency of the variance estimator is not proved, including in the rebuttal, which simply says "bootstrap estimators of variance is consistent under certain regularity conditions". This is not a proof--what are these conditions, and what papers prove that consistency under those conditions?
> >
> > W2: Thank you.
> >
> > W3: Consistency to 0 of the debiased lasso does not imply asymptotic symmetry of the null knockoff statistics. E.g., a Gaussian with mean 1/n and standard deviation 1/n converges to 0 but is not asymptotically symmetric about 0.
> >
> > As these soundness issues (primarily W1+W3) were the main reason for my low score, I am not revising my score at this time. I generally find the task-agnostic argument compelling, though think this could have been communicated better in the paper. The runtime experiment and discussion is great, but this needs to be in the paper (and the authors have not indicated that they will add it to the paper).

---

> ### Author Response · Authors · 2024-08-06
>
> **Q (continued):**
> * Even for M-estimation tasks that have already been implemented, PSPS offers the additional advantage of relying solely on summary statistics. The “task-specific derivations” mentioned throughout our paper were not only referring to statistical tasks, but also scientific tasks. Real-world data analysis in any scientific discipline often involves conventions and nuisances that require careful consideration. For example, our work is partly motivated from genome-wide association studies (GWAS). Statistically, GWAS is a linear regression that regresses an outcome (e.g., height) on many genetic variants. While the regression-based statistical foundation is simple, conducting a valid GWAS requires accounting for numerous technical issues, such as sample relatedness (i.e., study participants may be genetically related) and population structure (i.e., unrelated individuals of the same ancestry are both genetically and phenotypically similar, creating confounded associations in GWAS). Sophisticated algorithms and software have been developed to address these complex issues (`Mbatchou et al. 2021`). It will be very challenging if all these important features need to be reimplemented in an ML-assisted GWAS framework. With our PSPS protocol, researchers can utilize existing algorithms and software that are highly optimized for genetic applications to perform ML-assisted GWAS. This adaptability is not just limited to GWAS but is a major feature of our approach across scientific domains. A main result of this paper is that PSPS enables researchers to conduct ML-assisted inference using well-established data analysis pipelines.
>
> **L:**
> Following this suggestion, we have conducted experiments to compare the computational efficiency of our method (PSPS) against existing methods. Utilizing a dataset with 500 labeled and 10,000 unlabeled data points, PSPS required 1.62 seconds for linear regression and 8.27 seconds for logistic regression using 200 bootstrap resampling. The computation of one-step de-biasing using summary statistics alone took 0.032 seconds for linear regression and 0.033 seconds for logistic regression. Current methods, which estimate asymptotic variance via the closed form derived by the Central Limit Theorem instead of resampling, ranged from 0.024 to 0.049 seconds for linear regression and 0.032 to 0.077 seconds for logistic regression. Although PSPS is slower due to its resampling nature, the overall runtime remains relatively short.
>
> | Method | Linear regression  | Logistic regression |
> | ------------- | ------------- |------------- |
> | PSPS | 1.62s | 8.27s |
> | PPI  | 0.024s  | 0.032s |
> | PPI++  | 0.031s  | 0.077s |
> | POP-Inf  | 0.049s |0.034s |
>
> We also note that we designed PSPS to utilize summary statistics, aiming to integrate seamlessly with existing computationally efficient software routinely used in data analysis. For example, Regenie is a software employed in GWAS that allows for fast computation of tens of millions of linear regressions using tens of thousands of samples  (`Mbatchou et al. 2021`). Our protocol involves initially generating summary statistics using such a software tool, followed by their integration, allows high computational efficiency.
>
> **Final note**: We are excited that you find our work tackling an important problem and appreciate our theoretical study in statistical efficiency. If you have any further questions, please do not hesitate to let us know. If our responses have resolved your concerns, we kindly request you to consider increasing your score and championing our paper.

---

> ### Author Response · Authors · 2024-08-11
>
> Thank you for your valuable comments!
>
> **W1:** We have included in the manuscript the formal regularity conditions required for consistent bootstrap variance estimation. These conditions are detailed in Theorem 3.10 (i) from `Shao, Jun, and Dongsheng Tu. The jackknife and bootstrap. Springer Science & Business Media, 2012.`, which proves the consistency of the bootstrap variance estimator.
>
> Below is the detailed theorem:
>
> We assume that $X _1, \cdots, X _n$ are i.i.d random p-dimensional vectors from distribution $F$. Let $T _n = T _n(X _1, \cdots, X _n)$ be a n estimator of an unknown parameter $\theta$, and $\Re _n = \sqrt{n}(T _n - \theta) \sim N(0, \sigma_n^2)$. Let $\{X _1^*, \ldots, X _n^*\}$ be a bootstrap sample from the empirical distribution $F _n$ based on $X _1, \cdots, X _n$, $T_n^*=T_n\left(X_1^*, \ldots, X_n^*\right)$ and $\Re_n^*=\sqrt{n}\left(T_n^*-T_n\right)$. Denote the bootstrap variance estimator $v _{\text {boot}} = \text{Var}(T_n^*)$.
>
> Theorem 3.10 (i) Let $T _n=T\left(F _n\right)$, assume that $T$ is $\rho _{\infty}$-Fréchet differentiable at $F$, and
> $
> \max _{i_1, \ldots, i _n}\left|T _n\left(X _{i _1}, \ldots, X _{i _n}\right)-T _n\right| / \tau _n \rightarrow _{a . s .} 0
> $
> where the maximum is taken over all integers $i _1, \ldots, i _n$ satisfying $1 \leq$ $i _1 \leq \cdots \leq i _n \leq n$, and $\{ \tau _n \}$ is a sequence of positive numbers satisfying $\liminf _n \tau _n>0$ and $\tau _n=O\left(e^{n^q}\right)$ with a $q \in (0, \frac{1}{2})$., then $v _{\mathrm{boot}} / \sigma _n^2 \rightarrow _p 1$, where $\sigma _n^2=n^{-1} E[\phi _F(X _1)]^2>0$
>
> We have also added a remark in our paper to refer the reader to the theoretical results in a more recent paper `Hahn, Jinyong, and Zhipeng Liao. "Bootstrap standard error estimates and inference." Econometrica 89.4 (2021): 1963-1977.` for bootstrap variance estimation. This paper (THEOREM 1) shows that common bootstrap based standard error in fact leads to a valid but (potentially conservative) inference.
>
> **W3:** Thank you for bringing this to our attention. Since we have decided to remove Section 4 from the paper and keep only the remarks in Section 3, we will no longer include the theoretical results related to this section in the paper. Instead, we will empirically verify the performance of ML-assisted FDR control (as we have done in Section 5 of the paper). As the reviewer pointed out, this does not affect the main contribution of our paper, which is the compelling feature of "task-agnostic" ML-assisted inference.
>
> **Task-agnostic feature of PSPS:**
> In Section 3.1, we have added a paragraph specifically highlighting why PSPS is considered "task-agnostic." This addition includes the three bullet points from our previous rebuttal that clearly delineate the relevant scenarios.
>
> **Runtime comparison:**
> We have added the runtime comparison into the Section 5 (Numerical experiments and real data application) and the relevant discussions in the Section 6 (Conclusion) in the paper.
>
> We hope our responses address your concerns, and please do not hesitate to let us know if you have any further questions.

---

> > ### Comment · Reviewer_DsrD · 2024-08-11
> >
> > Thank you, this does address my main soundness concerns, and I am raising my score accordingly.

---

### Official Review · Reviewer_hhpT · 2024-07-13

**Soundness:** 3
**Presentation:** 3
**Contribution:** 3
**Rating:** 7
**Confidence:** 1

**Summary:**

The paper proposes a novel statistical framework for ML-assisted inference. It describes how labeled data, together with unlabeled data and a pre-trained ML model, can be used for statistical inference. The paper establishes the asymptotic properties and optimality of the framework and then evaluates it empirically on simulated and real datasets.

**Strengths:**

- The paper is well-written.
- Framework is flexible. It can work with almost any established data analysis routine and machine learning model.
- The framework is justified both theoretically and empirically.

**Weaknesses:**

- The limitations could have been discussed more thoroughly.

**Questions:**

- What are the limitations of the proposed framework?

**Limitations:**

The authors state that the limitations are discussed in the Conclusion section of the paper, but it seems they are not discussed there.

---

> ### Author Rebuttal · Authors · 2024-08-06
>
> **Weaknesses, Questions, and Limitations:**
>
> Thank you for the suggestion. One limitation is the computational burden of the naive bootstrap approach. In our original submission, we discussed the future direction of improving the speed of resampling. In the revised manuscript, we have conducted additional experiments to compare the computational efficiency of PSPS with existing methods.  Utilizing a dataset with 500 labeled and 10,000 unlabeled data points, PSPS required 1.62 seconds for linear regression and 8.27 seconds for logistic regression using 200 bootstrap resampling. The computation of one-step de-biasing using summary statistics alone took 0.032 seconds for linear regression and 0.033 seconds for logistic regression. Current methods, which estimate asymptotic variance via the closed form derived by the Central Limit Theorem instead of resampling, ranged from 0.024 to 0.049 seconds for linear regression and 0.032 to 0.077 seconds for logistic regression.
>
> | Method | Linear regression  | Logistic regression |
> | ------------- | ------------- |------------- |
> | PSPS | 1.62s | 8.27s |
> | PPI  | 0.024s  | 0.032s |
> | PPI++  | 0.031s  | 0.077s |
> | POP-Inf  | 0.049s |0.034s |
>
> These experiments demonstrate that while PSPS is slower than current methods due to its reliance on resampling to estimate the variance, the overall speed remains reasonable and fast. One potential solution to further improve speed could involve adopting more advanced methods for faster resampling such as Bag of Little Bootstraps (`Kleiner et al. 2014`). We have discussed this more thoroughly in the manuscript.
>
> **Final note**: We are excited that you find our method flexible and justified with both theoretical and empirical analysis. If you have any further questions, please do not hesitate to let us know. If our responses have resolved your concerns, we kindly request you to consider increasing your score and championing our paper.

---

> > ### Comment · Reviewer_hhpT · 2024-08-08
> >
> > Thank you for your response. I appreciate the additional experiments and the discussion about the proposed framework's limitations.

---

### Official Review · Reviewer_TkNf · 2024-07-13

**Soundness:** 3
**Presentation:** 3
**Contribution:** 3
**Rating:** 7
**Confidence:** 4

**Summary:**

The paper introduces a task-agnostic approach to inference with machine learning predictions. The basic idea follows a similar recipe as prediction-powered inference and related recent papers, but it makes use of resampling instead of the CLT with a plug-in estimate of the asymptotic covariance to avoid relying on analytical problem-specific derivations and expressions.

**Strengths:**

The paper effectively demonstrates the applicability of the method beyond M-estimation (which is what most previous papers have focused on), giving several important applications. The method is simple and elegant. The real-data application is very compelling.

**Weaknesses:**

I agree with your point that the recent methods require task-specific derivations. However, it's worth noting that, since for M-estimators we know we get asymptotic normality, we can use the same estimators but instead of deriving the asymptotic variance through the CLT we can use resampling to estimate the asymptotic variance. This is just to say that for M-estimators we can use the old estimators and get inference without problem-specific derivations. (Your other criticisms still apply.)

In L144 it says that resampling-based inference focuses on bias and variance estimation. I don't really agree with this. Plenty of resampling-based inference focuses on type I error control (confidence intervals and p-values).

The mean-estimation result in Section 2.2 is completely borrowed from prior work. Please make this clear. Otherwise it looks like these are new results in this paper.

In Algorithm 1 it is unclear what the actual output is. Please write it out using mathematical symbols.

Regarding Proposition 2, I would say that the claim looks a bit too strong. In PPI++ the authors state that they give the example with minimizing the trace as an example, but that other scalarizations of the covariance are clearly possible. Your Proposition 2 just chooses a different scalarization for the comparison so it is clearly better than the trace example from PPI++. But the PPI++ argument would give the same asymptotic variance under your scalarization.

It would be helpful to give an example of when the p-values in Proposition 3 will be PRDS.

Some stylistic comments:
- The grammatically correct way to spell the title would be "Task-Agnostic Machine-Learning-Assisted Inference."
- in abstract: constraints -> constrains
- Very often throughout the paper you say "standard error" but you write the variance. For example, see L68 or L73. Please be consistent: either say variance and write Var(...) or write out the standard error.
- In L75, you are clearly describing a particular class of ML-assisted inference methods. Not every method that is ML-assisted follows that description. Add references so it's clear what you are referring to.
- L91: motivated the observation -> motivated by the observation
- L99: inputs -> inputting
- There are other typos and minor stylistic issues. Please go through the paper carefully.

**Questions:**

- Can you elaborate on the point about the privacy-preserving feature of your method? I didn't understand the main point. A particular use case would be helpful.
- I'm surprised that PSPS and PPI++ are not getting the same interval widths in linear and logistic regression in Figure 2. Is it because you are tuning PPI++ for the trace objective, as discussed above? The two methods (assuming the right tuning) should have exactly the same asymptotics in this problem. This should also be made clear in words.
- I'm curious about the details behind the real-data application. Can you elaborate on how you computed the predictions and how you used cross-validation to avoid overfitting?

**Limitations:**

I don't think a further discussion is necessary.

---

> ### Author Rebuttal · Authors · 2024-08-06
>
> **W1**: We first want to highlight that current methods and their implementations typically estimate asymptotic variance using the CLT rather than through resampling (for example, `Angelopoulos et al., 2023` PPI and PPI++). In addition, while resampling-based approaches can bypass the derivation of asymptotic variance, task-specific derivations for the loss function of a new ML-assisted M-estimator are still essential to obtaining the point estimator, which precedes resampling-based estimation of its uncertainty.
>
> We also want to add that the “task-specific derivations” mentioned throughout our paper were not only referring to statistical tasks, but also scientific tasks. Real-world data analysis in any scientific discipline often involves conventions and nuisances that require careful consideration. For example, our work is partly motivated from genome-wide association studies (GWAS). Statistically, GWAS is a linear regression that regresses an outcome (e.g., height) on many genetic variants. While the regression-based statistical foundation is simple, conducting a valid GWAS requires accounting for numerous technical issues, such as sample relatedness (i.e., study participants may be genetically related) and population structure (i.e., unrelated individuals of the same ancestry are both genetically and phenotypically similar, creating confounded associations in GWAS). Sophisticated algorithms and software have been developed to address these complex issues (`Mbatchou et al. 2021`). It will be very challenging if all these important features need to be reimplemented in an ML-assisted GWAS framework. With our PSPS protocol, researchers can utilize existing algorithms and software that are highly optimized for genetic applications to perform ML-assisted GWAS. This adaptability is not just limited to GWAS but is a major feature of our approach across scientific domains. A main result of this paper is that PSPS enables researchers to conduct ML-assisted inference using well-established data analysis pipelines.
>
> **W2**: We have revised the texts to “whereas resampling-based inference focuses on bias and variance estimation, and type-I error control.”.
>
> **W3**: We have added relevant citations when we discuss the mean-estimation result in Section 2.2.
>
> **W4**: We have revised Algorithm 1 following the suggestion.
>
> **W5**: We would like to clarify that the asymptotic variance of PSPS will always be lower than that of PPI++, irrespective of the scalarization method used. The reason is that the weights matrix in PPI++ is a special case of the more general weights matrix used in PSPS, and we have demonstrated in Proposition 2 that the weighting matrix in PSPS is optimal. The weighting matrix $\omega_0$ in PSPS is a Q×Q matrix with no constraint, where Q is the number of dimensions for the parameters. In contrast, PPI++ constrains its matrix to be diagonal with the same diagonal elements. Therefore, PSPS enables information sharing across different parameter coordinates, enhancing estimation precision. The choice of the weighting matrix in PSPS also facilitates element-wise variance reduction: each diagonal element of the variance-covariance matrix is reduced. In contrast, the single parameter in PPI++ can only target overall trace reduction or variance reduction of a specific element.
>
> To provide further intuition, we consider a linear regression with two predictors:  $Y \sim \theta_1 X_1 + \theta_2 X_2$. The summary statistics for PSPS can be expressed as: $[\hat{\theta} _{1L}, \hat{\theta} _{2L}, \hat{\eta} _{1L}, \hat{\eta} _{2L}, \hat{\eta} _{1U}, \hat{\eta} _{2U}]^T$. For PSPS, since $\hat{\theta} _{\text{PSPS}}= \begin{bmatrix}\hat{\theta} _{1L} \\\ \hat{\theta} _{2L} \end{bmatrix} - \begin{bmatrix}w _1 & w _{12} \\\ w _{12} & w _2 \end{bmatrix}\begin{bmatrix}\hat{\eta} _{1L} \\\ \hat{\eta} _{2L} \end{bmatrix} + \begin{bmatrix}w _1 & w _{12} \\\ w _{12} & w _2 \end{bmatrix}\begin{bmatrix}\hat{\eta} _{1U} \\\ \hat{\eta} _{2U} \end{bmatrix}$, the final estimator for $\theta_1$ is $\hat{\theta} _{\text{PSPS}, 1} = \hat{\theta} _{1L}-w _1 \hat{\eta} _{1L}+w _1 \hat{\eta} _{1 U}-\mathrm{w} _{12} \hat{\eta} _{2L}+\omega _{12} \hat{\eta} _{2U}$.
>
> In comparison, since $\hat{\theta} _{\text{PPI++}}= \begin{bmatrix}\hat{\theta} _{1L} \\\ \hat{\theta} _{2L} \end{bmatrix} - \begin{bmatrix}w & 0 \\\ 0 & w \end{bmatrix}\begin{bmatrix}\hat{\eta} _{1L} \\\ \hat{\eta} _{2L} \end{bmatrix} + \begin{bmatrix}w & 0 \\\ 0  & w \end{bmatrix}\begin{bmatrix}\hat{\eta} _{1U} \\\ \hat{\eta} _{2U} \end{bmatrix}$, its estimator for $\theta_1$ is $\hat{\theta} _{\text{PPI}++, 1} = \hat{\theta} _{1L}-w \hat{\eta} _{1L}+w \hat{\eta} _{1U}$.
>
> Since $\hat{\theta} _{\text{PSPS}, 1}$ involves two zero-augmentation terms (i.e., $-w _1 \hat{\eta} _{1L}+w _1 \hat{\eta} _{1 U}$ and $-\mathrm{w} _{12} \hat{\eta} _{2L}+\omega _{12} \hat{\eta} _{2U}$ ), its asymptotic variance should be less than or equal to that of PPI++ with one augmentation term. Therefore, PSPS borrows information from both coordinates, but PPI++ is restricted to information from only the first coordinate. Although the PPI++ can be used under a different scalarization, it still contains one augmentation term. Only in a one-dimensional parameter estimation task, PPI++ and PSPS will have the same asymptotic variance. We have clarified this in the revised manuscript. We have also added the example above and an example for a one-dimensional parameter estimation task to clearly state the difference between PSPS and PPI++.
>
> **Due to the character limit, we have placed our response to the remaining concerns in the comment section of the review.**

---

> ### Author Response · Authors · 2024-08-06
>
> **W6:** Examples of PRDS p-values include independent p-values and p-values from test statistics that are jointly normally distributed, if all correlations between test statistics are positive (`Wang, Ruodu, and Aaditya Ramdas. "False discovery rate control with e-values." Journal of the Royal Statistical Society Series B: Statistical Methodology 84.3 (2022): 822-852.`).
>
> **W7:** We appreciate these suggestions. We have thoroughly addressed these issues in the revised manuscript.
>
> **Q1:** The “privacy-preserving feature” of PSPS refers to the fact that we only require summary statistics as input for inference, rather than individual-level raw data (features $X$ and label $Y$). This terminology is commonly used in human genetics, healthcare, and multicenter electronic health record analysis to describe methods that enable statistical inference without directly accessing personal data. This approach is analogous to federated learning (`Kairouz, Peter, et al. 2021`) in the machine learning literature. For example, consider a scenario where labeled data is in one center and unlabeled data in another, yet researchers cannot access individual-level data from both centers simultaneously. Under such conditions, current ML-assisted inference, which relies on accessing both labeled and unlabeled data to minimize a joint loss function, is not feasible. However, PSPS circumvents this issue by aggregating summary statistics from multiple centers, thus performing statistical inference while upholding the privacy of individual-level data.
>
> We also acknowledge that in the machine learning literature, "privacy-preserving" often specifically refers to techniques like differential privacy. To avoid confusion, we have revised the terminology from “privacy-preserving” to "federated inference" and provided an example to illustrate this use case more clearly.
>
> **Q2:** First, we apologize for the error that we mistakenly used Figure 2e for Figure 2f due to a typo in the code for making the figure. The correct Figure 2f is attached in the rebuttal pdf. However, this does not change our main results related to statistical efficiency: PSPS is more efficient than PPI++ and other existing methods. As we have previously explained in our comparison of PSPS and PPI++, PSPS leverages information across multiple coordinates in a multi-dimensional parameter estimation task, resulting in asymptotic variances that are less than or equal to those produced by PPI++. Regarding the concern about the choice of tuning, the POP-Inf method, which tunes the element-wise variance, also shows wider confidence intervals than PSPS in linear regression.
>
> **Q3:** Our prediction pipeline comprises two components: prediction for unlabeled data and prediction for labeled data. To predict bone mineral density in unlabeled data, we first selected predictive features by 1) calculating the correlation of bone mineral density with 466 other variables (sample size > 200,000 from UK Biobank) using labeled data and 2) selecting the top 50 variables with the highest correlations as inputs for the SoftImpute algorithm to predict bone mineral density in the unlabeled data.
>
> For the labeled data, we employ a similar approach but incorporate 10-fold cross-validation to prevent overfitting. We select the predictive variables and train the SoftImpute model using 90% of the labeled data. We then perform predictions on the remaining 10% in each fold and repeat this process 10 times across all folds.
>
> We have included these details in the Appendix of the manuscript.
>
> **Final note**: We are excited that you find our method flexible, simple and elegant, and the real data application compelling. If you have any further questions, please do not hesitate to let us know. If our responses have resolved your concerns, we kindly request you to consider increasing your score and championing our paper.

---

> > ### Comment · Reviewer_TkNf · 2024-08-11
> >
> > Thank you for the response! It was very helpful and clarifying.

---

### Author Rebuttal · Authors · 2024-08-06

We thank the reviewers for providing valuable suggestions that helped us improve our paper. We are particularly encouraged that the reviewers have found that (i) the problem we study in this paper is important (R-DsrD), (ii) our method is simple and elegant (R- TkNf), (iii) our flexible statistical framework can be applied to almost any established data analysis routine (R-hhpT and R-Tknf) and machine learning model (R-hhpT), (iv) our method is justified both theoretically and empirically (R-hhpT), and the real-data application is very compelling (R-Tknf), (v) our method dominates existing approaches in terms of statistical efficiency (R-DsrD), (vi) the paper provides a well-rounded analysis on the proposed method with sensitivity analysis on distributional shifts (R-5S3y).

In response to the feedback, we have addressed each concern, added new experimental results and clarification, and updated our paper accordingly.

A summary of our major changes is provided below.
1. We have explained the "task-agnostic" feature of our method.
2. We have clarified the "privacy-preserving" aspects of our method.
3. We have elucidated the connection between our ML-assisted statistical inference and semi-supervised learning in the machine learning literature.
4. We have addressed technical issues related to variance estimation in Theorem 1 and corrected typos throughout the paper.
5. We have included examples to illustrate how our method outperforms existing approaches in terms of statistical efficiency.

A detailed point-by-point response to each reviewer's comments is available in the rebuttal section corresponding to each reviewer.

---

### Decision · Program_Chairs · 2024-09-25

**Decision:**

Accept (poster)

**Comment:**

The paper proposes a framework for statistical inference that combines labeled, unlabeled through a post-inference solution that can be used with different data analysis routines. The reviewers unanimously recommend its acceptance. They noted that the proposed method is simple and flexible, the evaluations were comprehensive through theoretical and empirical analysis and highlighted its application to important areas. I therefore recommend its acceptance.